# Adsorption separation of heavier isotope gases in subnanometer carbon pores

Sanjeev Kumar Ujjain [1], Abhishek Bagusetty [2], Yuki Matsuda[3], Hideki Tanaka [1], Preety Ahuja [1], Carla de Tomas [4], Motomu Sakai [5], Fernando Vallejos-Burgos [1,6], Ryusuke Futamura [1], Irene Suarez-Martinez[4], Masahiko Matsukata[7], Akio Kodama[3], Giovanni Garberoglio [8,9], Yury Gogotsi[1,10], J. Karl Johnson [2] & Katsumi Kaneko [1✉]

Isotopes of heavier gases including carbon ($^{13}C/^{14}C$), nitrogen ($^{13}N$), and oxygen ($^{18}O$) are highly important because they can be substituted for naturally occurring atoms without significantly perturbing the biochemical properties of the radiolabelled parent molecules. These labelled molecules are employed in clinical radiopharmaceuticals, in studies of brain disease and as imaging probes for advanced medical imaging techniques such as positron-emission tomography (PET). Established distillation-based isotope gas separation methods have a separation factor ($S$) below 1.05 and incur very high operating costs due to high energy consumption and long processing times, highlighting the need for new separation technologies. Here, we show a rapid and highly selective adsorption-based separation of $^{18}O_2$ from $^{16}O_2$ with $S$ above 60 using nanoporous adsorbents operating near the boiling point of methane (112 K), which is accessible through cryogenic liquefied-natural-gas technology. A collective-nuclear-quantum effect difference between the ordered $^{18}O_2$ and $^{16}O_2$ molecular assemblies confined in subnanometer pores can explain the observed equilibrium separation and is applicable to other isotopic gases.

[1] Research Initiative for Supra-Materials, Shinshu University, Nagano City, Japan. [2] Department of Chemical & Petroleum Engineering, University of Pittsburgh, Pittsburgh, PA, USA. [3] Institute of Science and Engineering, Kanazawa University, Kanazawa, Japan. [4] Department of Physics and Astronomy, Curtin University, Perth, WA, Australia. [5] Research Organization for Nano and Life Innovation, Waseda University, Tokyo, Japan. [6] Morgan Advanced Materials, Carbon Science Centre of Excellence, State College, PA, USA. [7] School of Advanced Science and Engineering, Waseda University, Tokyo, Japan. [8] European Centre for Theoretical Studies in Nuclear Physics and Related Areas (FBK-ECT*), Strada delle Tabarelle 286, I-38123 Trento, Italy. [9] Trento Institute for Fundamental Physics and Applications (TIFPA-INFN), via Sommarive 18, I-38213 Trento, Italy. [10] Department of Material Science and Engineering, and A.J. Drexel Nanomaterials Institute, Drexel University, Philadelphia, PA, USA. ✉email: kkaneko@shinshu-u.ac.jp

Stable isotope technologies have attracted increasing interest over the last decade due to their indispensable role in plant, aquatic, and animal life[1–4]. They are widely used as radiolabelled probes in isotopic analysis methods[5–8], therapeutic cancer drugs[9,10], and environmental studies[11,12]. Naturally occurring heavier gas isotopes including carbon ($^{13}C$ and other isotopes), nitrogen ($^{13}N$), and oxygen ($^{18}O$) are extensively used for such radiolabeling because their molecular weight difference from their main isotope is quite minor compared to the lighter hydrogen ($^{1}H$)-deuterium ($^{2}H$) molecules, resulting in a lower isotopic effect and hence only a minor perturbation to the biochemical behavior[1,2,13]. The positron ($\beta^+$)-emitting isotopes, including $^{15}C/^{11}C$, $^{13}N$, and $^{18}F$, can be used as imaging probes in the radionuclide-based molecular imaging positron emission tomography (PET) technique and other useful clinical radiopharmaceuticals[2]. In particular, $^{18}O_2$ acts as a target in cyclotrons for producing [$^{18}F$]-labeled 2-deoxyglucose (FDG) as a $\beta^+$-emitting radioisotope for PET biomedical imaging. Of all of the $\beta^+$ emitters employed in PET, the $^{18}F$ isotope is the preferred probe in radiopharmaceuticals because of its relatively long half-life (110 min), which allows its use in clinical PET imaging centers that lack radiochemistry facilities, and its low positron energy (0.64 MeV) that enhances the spatial resolution to produce optimal physical characteristics[2]. Other $\beta^+$ emitters involving $^{11}C$ or $^{13}N$ exhibit short half-lives and can only be utilized at imaging centers with a cyclotron and radiochemistry facility[2]. PET imaging can show abnormalities that cannot be detected by other techniques, resulting in the early and more accurate diagnosis of infected or cancerous tissues[14]. Furthermore, $^{18}O_2$ can be introduced into the body by simple inhalation to study the function of the brain for the treatment of patients with diseases, such as schizophrenia, manic depression[2,10,15,16]. Therefore, enhanced production of $^{18}O$ is indispensable for achieving improvements in healthcare, environmental protection, and basic sciences.

The natural abundance of oxygen isotope $^{18}O$ is very low and is only 0.204 at.% compared to the principal $^{16}O$ isotope (99.76 at. %). Isotopic separation is currently only possible with a limited number of techniques, such as cryogenic distillation of oxygen/ nitric oxide/water[17,18], membrane distillation[19,20], isotope exchange reaction[21], and thermal diffusion[22]. These methods require complex equipment and are time and energy-intensive. Consequently, alternative technologies are needed to meet the high demand for $^{18}O$ in healthcare and environmental science fields.

Recently, low-temperature adsorption-based separation enabled by the differences in isotope adsorption in nanopores due to the higher zero-point energy of translational motion of light isotope molecules (so-called quantum molecular sieving, QMS) has been demonstrated[23,24]. Consequently, QMS is the most promising alternative to the currently used methods for the separation of light isotopes such as hydrogen or helium[25,26]. However, the effective size differences for heavier gas isotopes such as $^{18}O_2$ and $^{16}O_2$ or $^{13}CH_4$ and $^{12}CH_4$ at 112 K, as estimated from the thermal de Broglie wavelength for the QMS effect, are only 0.0017 nm and 0.0012 nm, respectively. In addition, the Feynman–Hibbs potential calculation shows a difference of only 0.07 K for both isotope gases (Supplementary Fig. 1 and Supplementary Note 1), showing that QMS is ineffective for heavier gas isotopes. Therefore, we cannot expect that efficient separation of oxygen, methane, or other heavier gas isotopes can be achieved with nanoporous materials through the QMS mechanism.

The $O_2$ or $CH_4$ molecule has other vibrational and rotational degrees of freedom that depend on the molecular weight. The vibrational states are essentially the same in the gas and adsorbed phases and cannot lead to the selective adsorption of $^{18}O_2$ or $^{13}CH_4$. By contrast, the rotational degrees of freedom of the

molecules adsorbed in the nanopores are highly restricted below 130 K, unlike those of gaseous molecules[27]. Nevertheless, a very small difference in the rotational energies of these isotope molecules coupled with translational motion[28] is not sufficient to induce the selective adsorption of $^{18}O_2$ or $^{13}CH_4$. However, when $O_2$ or $CH_4$ molecules are densely packed at quasi-solid densities in nanopores at low temperature, collective quantum motion associated with vibrational and rotational modes can give rise to a sufficiently large adsorption-energy difference for oxygen and methane isotope molecules to enable the selective separation, as described in this article.

In this work, we report an industrially applicable adsorption-separation route of $^{18}O_2$ from a $^{16}O_2–^{18}O_2$ mixture using nanoporous solids at ~112 K, utilizing the cryogenic facilities of liquefied natural gas industries to save energy. The observed dynamic selectivity or separation factor $S(^{18}O_2/^{16}O_2)$ exceeds 60, depending on the nanoporous adsorbent, as is further confirmed by breakthrough curve experiments. We also demonstrate an equally successful separation of $^{13}CH_4/^{12}CH_4$ under similar conditions.

## Results

**Dynamic adsorption separation of CDC.** The dynamic adsorption-based separation was measured at different temperatures using a custom-made flow-type mixed gas adsorption apparatus coupled to a quadrupole mass spectrometer (MS) (Supplementary Figs. 2 and 3; Supplementary Notes 2 and 3)[29]. High purity $^{16}O_2$ (99.99995%) and $^{18}O_2$ (≥98%) gases were mixed to achieve different compositions ($^{18}O_2$% varied from 4.8 to 70 at. %) and then were introduced into the sample cell at a constant temperature. Similarly, $^{12}CH_4$ (99.9999%) and $^{13}CH_4$ (99%) were used to prepare isotopic mixtures. The adsorbents employed were carbide-derived carbon (CDC) produced by the chlorination of TiC[30] and activated carbon fibers (ACFs) with slit-shaped pores. The effective pore widths determined from the $N_2$ adsorption isotherms at 77 K are 0.7 nm for CDC and ACF5 and 0.8 nm and 1.1 nm for ACF10 and ACF20, respectively (Supplementary Figs. 4–6 and Supplementary Table 1). For theoretical calculations, the pore widths were also converted into the effective pore widths by subtracting 0.334 nm (the average graphite interlayer spacing) from the actual diameters[31]. We also considered open single-walled carbon nanotubes ($SWCNT_{ox}$) with cylindrical 1.5-nm and 1.0-nm wide pores and zeolites MS4A, MS5A, MFI, and BEA with interconnected cylindrical pores with the widths 0.40, 0.50, 0.55, and 0.66 nm, respectively; these materials were pre-evacuated prior to the dynamic and equilibrium adsorption measurements.

The adsorption selectivity $S$ for $^{18}O_2$ against $^{16}O_2$ is defined as:

$$S\left(^{18}O_2/^{16}O_2\right)_{(ads-g)} = \frac{\left(^{18}O_2/^{16}O_2\right)_{ads}}{\left(^{18}O_2/^{16}O_2\right)_g} \quad (1)$$

where $(^{18}O_2/^{16}O_2)_{ads}$ are the mole fractions of $^{18}O_2$ and $^{16}O_2$ in the adsorbed phase and $(^{18}O_2/^{16}O_2)_g$ are the mole fractions of $^{18}O_2$ and $^{16}O_2$ in the bulk gas phase, respectively. For simplicity, $S$ $(^{18}O_2/^{16}O_2)_{(ads-g)}$ hereafter will be denoted as $S$.

Figure 1 illustrates that our cryogenic adsorption separation is much simpler, more energy-efficient, and compact than traditional cryogenic distillation involving sophisticated equipment with a distillation column with a height of tens of meters. For the selectivity observed from the present adsorption separation, only three adsorption stages are estimated to be required to obtain >95% purity of $^{18}O_2$. The cryogenic adsorption-separation method utilizes nanoporous adsorbents that can selectively adsorb $^{18}O_2$ in a gaseous $^{18}O_2–^{16}O_2$ mixture as shown in Fig. 2a–d; the details of the selectivity calculations are presented

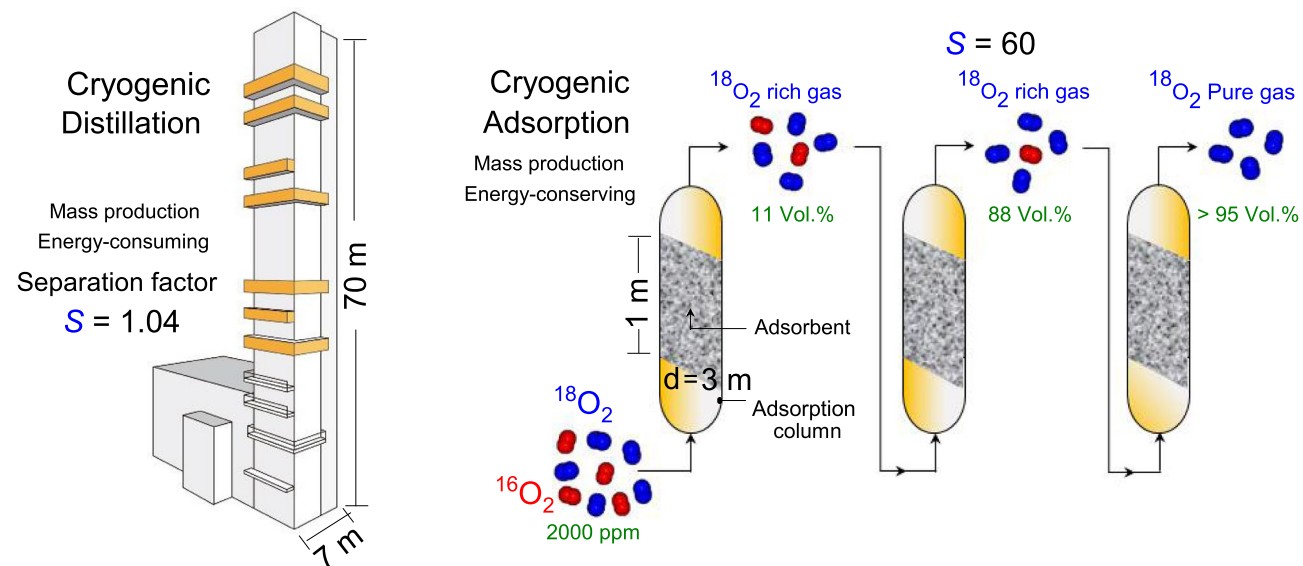

**Fig. 1 Cryogenic distillation vs. Cryogenic adsorption-based selective separation of oxygen isotopes using carbide-derived carbon (CDC).** The illustrative model quantitatively comparing the cryogenic distillation separation setup with the cryogenic adsorption-separation method. While distillation towers must be oriented vertically, adsorption beds can be configured in many different orientations. The nanoporous adsorbent bed in the adsorption column preferentially adsorbs $^{18}O_2$ according to the ratio determined by the adsorbent selectivity. The plausible adsorption-separation tower size was estimated for future consideration assuming the ideal conditions with the selectivity remaining at 60 and the adsorption capacity of 15 mmol/g. The assumption of the adsorbent density of 500 kg/m$^3$ and space velocity = 10 min$^{-1}$ leads to three separation towers with the capacity of 3 m (diameter) × 1 m (height) producing $^{18}O_2$ of >95%.

in Supplementary Figs. 7 and 8. The CDC adsorbent packed cell at 112 K shows the excellent selective separation of $^{18}O_2$ from the $^{18}O_2$–$^{16}O_2$ gaseous mixture with different $^{18}O_2$ at.%. Surprisingly, the dynamic adsorption selectivity $S$ of CDC (Fig. 2a) is greater than 60 at the initial stage, irrespective of the $^{18}O_2$ concentration, and decreases to 1.6–1.2 after 40 min (Fig. 2a, inset). Supplementary Fig. 9 shows the reproducibility of the selectivity. This decrease in the selectivity may be due to the decreased availability of the adsorption sites over time. CDC preferentially adsorbs $^{18}O_2$ even from the mixed gas with a very low content of $^{18}O_2$ ($^{18}O_2$ 4.8%, $^{16}O_2$ 95.2%). However, the initial drop in $S$ with time depends on the $^{18}O_2$ concentration. A smaller $^{18}O_2$% leads to a slower decay of $S$ with time. The time for which $S > 2.5$ decreases with increasing $^{18}O_2$%, as shown in Fig. 2b. The mixed gas with lower $^{18}O_2$ content (4.8% and 10.5%) exhibits $S > 2.5$ for 28 ± 2 min, while higher $^{18}O_2$ concentrations show less satisfactory performance. This result suggests that high selectivity persists under a wide range of isotopic concentration ratios, indicating the applicability of CDC for the separation of a more $^{18}O_2$ diluted mixed feed gas. This phenomenon is crucial for the industrial separation of these isotopes because the atmospheric content of $^{18}O$ is very low. The high selectivity can be demonstrated by comparing the adsorbed amounts of $^{18}O_2$ and $^{16}O_2$ by CDC for the equimolar mixture of oxygen isotopes at 112 K (Supplementary Figs. 10 and 11). $^{18}O_2$ is preferentially adsorbed from the beginning, creating a concentration difference of 20 μmol g$^{-1}$ over 120 min, resulting in high adsorption selectivity. Notably, the difference in the adsorption amounts of $^{18}O_2$ and $^{16}O_2$ can be observed from the beginning of the experiment; however, the selectivity can only be analyzed when the pore volumes of CDC are sufficiently crowded to allow few $^{18}O_2$ molecules to reach the MS, as shown in Supplementary Fig. 7 (Supplementary Note 4). Here, the pore volume filling, which is the ratio of the adsorbed $O_2$ volume derived from the liquid density of $O_2$ compared with the measured nanopore volume, is 2% for $^{18}O_2$; the expression for the adsorption by the pore filling is convenient for adsorption in

nanopores. Moreover, the pore volume filling of the CDC depends on various factors, such as the amount of mixture gas introduced, dosing rate of introduction, and temperature. If an adsorption experiment is performed with more mixed gas at a higher dosing rate, then the pore volume filling % will be higher, as demonstrated in Supplementary Fig. 12. The high selectivity trend in the initial stage is retained in the temperature range of 100–150 K, as shown in Fig. 2c. However, the $S$ at 1% pore volume filling and the time period for which $S > 2.5$ both decrease with increasing temperature (Fig. 2d). In addition, the pore volume filling of the mixed gas is larger at lower temperatures, and at 90 min, it decreases from 2.5% to 0.25% as the temperature increases from 100 K to 140 K (Supplementary Fig. 13). Consequently, the highly selective separation of $^{18}O_2$ from the mixed gas ($^{18}O_2 + {}^{16}O_2$) with a CDC of ~112 K is highly promising for the design of an efficient separation process.

The cryogenic adsorption separation demonstrates that $S > 60$ in the initial stage of adsorption (~1 min) arises from the selective adsorption of $^{18}O_2$ in the narrow sites of the pores (<0.4 nm) present in the CDC structure[32]. In our previous work, atomistic models of CDCs were generated via molecular dynamics simulation and validated by experimental adsorption measurements[33]. A slice with a depth of 3.5 nm and a side of 10.4 nm of a representative CDC model containing realistic slit-shaped pores is shown in Fig. 2e, left panel. The CDC porosity was evaluated by filling the structure with nonoverlapping spheres according to the Gubbins method[34]. Rendering of the spheres within the CDC structure aids the visualization of the pores, as shown in the right panel, where each sphere is colored according to its diameter. Narrow pores with diameters <0.4 nm are identified by dashed white ovals, confirming the presence of preferential adsorption sites for the $^{18}O_2$ molecules in CDCs. Figure 2f shows the experimental $S$ values of CDC at 100 K and 112 K compared with those of the other separation methods reported in the literature. The high selectivity $S = 116 \pm 14$ at 100 K or $S = 60 \pm 15$ at 112 K is ascribed to the preferential adsorption of $^{18}O_2$

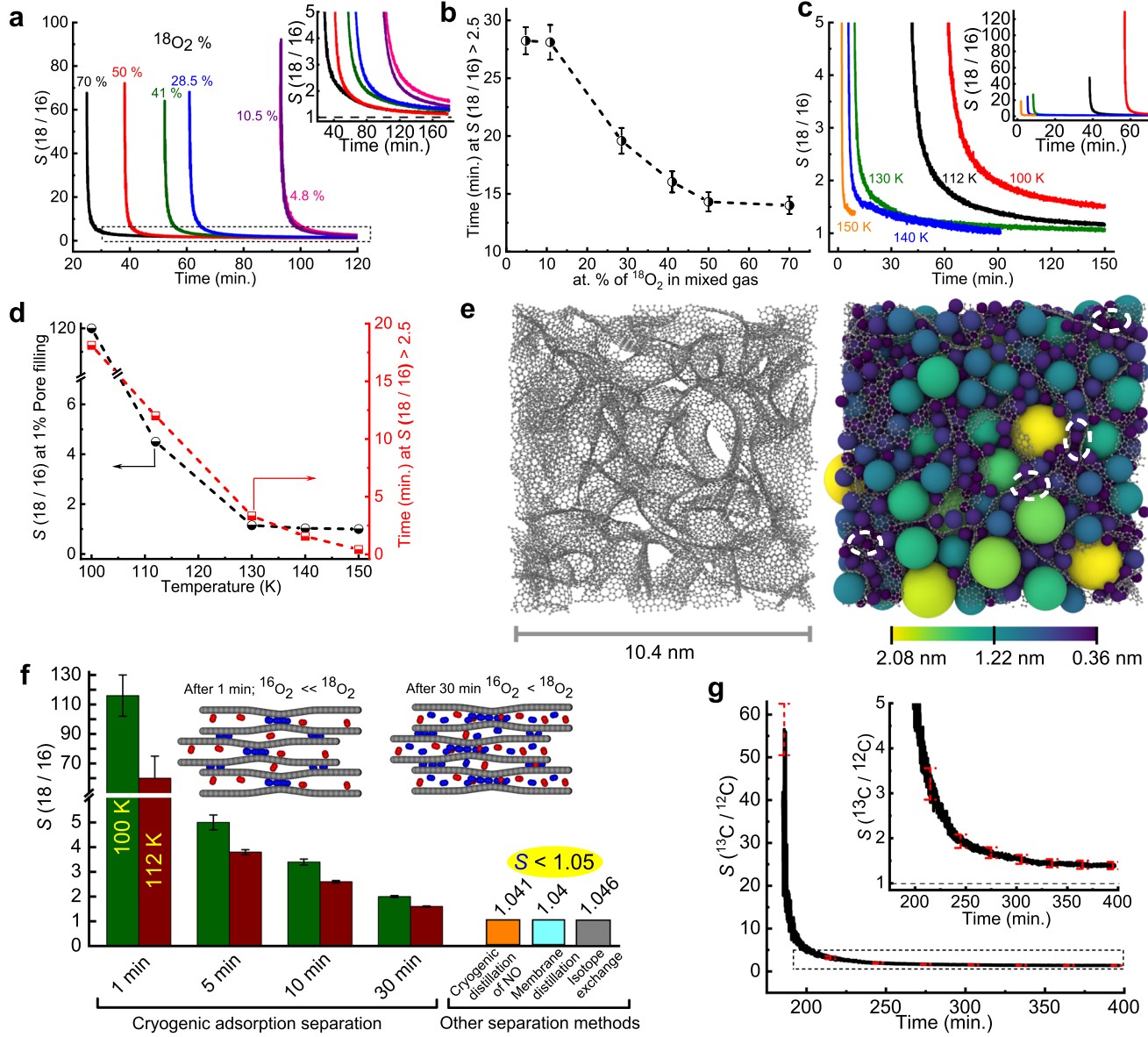

**Fig. 2 Kinetic cryogenic adsorption-based selective separation of oxygen and methane isotopes using carbide-derived carbon (CDC). a** $S(^{18}O_2/^{16}O_2)$ (referred to hereafter as S(18/16) in the figures) of the CDC at 112 K for different atomic % of $^{18}O_2$ in the feed gas mixture ($^{18}O_2 + ^{16}O_2$) under similar dosing rate (~1 mL min$^{-1}$). The percentage of $^{18}O_2$ in the feed gas mixture varied from 4.8 at.% to 70 at.%. The inset shows an enlarged view of the dotted regions. The black dashed line represents $S = 1$. **b** Time of $S > 2.5$ for different percentages of $^{18}O_2$ in the feed gas containing a mixture of the isotopes. Error bars (black lines) represent the standard deviations of three measurements. **c** Temperature dependence of selectivity $S$ examined for equimolar (50 at.% $^{18}O_2$) isotope mixtures using CDC at 100, 112, 130, 140, and 150 K. Inset shows a very high initial selectivity. **d** $S$ at 1% pore volume filling (black) and time for which $S > 2.5$ (red) for CDC as a function of temperature. **e**, left: Simulated CDC structure. Gray spheres and lines represent the carbon atoms and their bonds, respectively. The snapshot shows a slice of the structure with a length of 10.4 nm and a depth of 3.5 nm. Right: 3D rendering of the same slice filled with nonoverlapping spheres. The spheres are colored by diameter, with the values indicated in the color bar. Dashed white ovals identify narrow pore sites filled with spheres of diameter <0.4 nm. **f** Comparison of $S$ at different times at 100 K and 112 K for the CDC in this work with other separation methods from the literature. The inset shows illustrative models for the pore filling of CDC by $^{16}O_2$ and $^{18}O_2$ molecules after 1 min and 30 min. References correspond to cryogenic distillation (16,17), membrane distillation (18,19), and isotope exchange (20). **g** $S(^{13}CH_4/^{12}CH_4)$ (written as $S(^{13}C/^{12}C)$ in the figure) of CDC at 112 K for feed gas mixture ($^{13}CH_4 + ^{12}CH_4$) under similar dosing rate (~1 mL min$^{-1}$). The inset shows an enlarged view of the dotted regions. The error bar represented by the red dashed line is the standard deviations derived from three measurements.

molecules at the narrow pore sites (left inset in Fig. 2f). Even after 30 min, the CDC pores are filled with $S > 2$ (right inset in Fig. 2f), which is still twice that of the other separation methods. To further ascertain the observed high selectivity trends, adsorption–separation experiments are performed using CDC adsorbent for only 10, 15, and 20 min at 112 K. The mole fractions of the feed and adsorbed mixture isotope gas are

monitored using the time evolution of mass intensity. The molar ratio for the feed gas $^{18}O_2{:}^{16}O_2 = 1$. The observed mole fraction of the desorbed mixed gas is $^{18}O_2{:}^{16}O_2 = 1.4 \pm 0.005$ (17.5 Pa $^{18}O_2 + 12.5$ Pa $^{16}O_2$) after adsorption for 10 min. When adsorption was performed for 20 min, the desorbed mixed gas displayed $^{18}O_2{:}^{16}O_2 = 1.3 \pm 0.006$ (Supplementary Fig. 14). This shows that ~8% higher adsorption of $^{18}O_2$ occurs during the 10-min

selective adsorption experiment compared to the experiment carried out for 20 min, corresponding to a maximum difference of 1.33 mg/g between $^{18}O_2$ and $^{16}O_2$. These results confirmed that the initial high selectivity tends to decrease over time.

In addition, we have verified that this method is applicable to other gaseous heavy isotope pairs involving $^{13}CH_4$ and $^{12}CH_4$. The CDC demonstrates selective separation of $^{13}CH_4$ from the mixed gas ($^{13}CH_4 + ^{12}CH_4$) at 112 K with $S(^{13}C/^{12}C) = 56 \pm 6$ initially and maintains $S > 2.5$ for more than 40 min (Fig. 2g).

**Dynamic adsorption separation of different nanoporous adsorbents**. Other nanoporous materials with different pore geometries show similar selectivity trends even though their selective absorptivities are slightly inferior to that of CDC, as shown in Fig. 3a, b. The equilibration time depends on the pore aperture. ACF5 (slit pore width: 0.7 nm) shows high selectivity for an extended time interval with $S > 1.2$ up to 120 min (complete experiment time, see inset Fig. 3a). $SWCNT_{ox}$ with 1 nm cylindrical pore maintains $S > 1.2$ for more than 60 min, while zeolite MFI (pore width 0.55 nm) shows $S > 1.2$ for only 30 min (inset Fig. 3a, b). We have also measured the selectivity in AFC10, AFC20, $SWCNT_{ox}$ (1.5 nm), and zeolites BEA and MS5A, all of which exhibit lower selectivity. In addition, the adsorbents with pore widths of < 0.4 nm do not demonstrate significant selective separation performance (Supplementary Figs. 15 and 16). A significant difference in the adsorbed amounts of $^{18}O_2$ and $^{16}O_2$ can be observed in Fig. 3c, d. For ACF5, the adsorbed gas amount increases gradually, reaching a maximum adsorption difference of 13 $\mu mol\ g^{-1}$, while the corresponding values for $SWCNT_{ox}$ (1 nm) and zeolite MFI after 120 min are 10 $\mu mol\ g^{-1}$ and 16 $\mu mol\ g^{-1}$, respectively. The pore volume filling percent after 120 min varies from 0.6% for ACF20 (pore volume of 0.81 ml $g^{-1}$) to 5.2% for zeolite MFI (pore volume of 0.16 ml $g^{-1}$) (Supplementary Fig. 17). Figure 3e summarizes the $S$ values for all the nanoporous adsorbents at a pore volume filling of 1%, demonstrating the best selective adsorption behavior of CDC. This implies that the observed selectivity depends strongly on the pore size and the pore geometry (shape). The narrow pores present in the CDC structure have a 2D structure, while $SWCNT_{ox}$ (cylindrical pores) or zeolites (interconnected channeled cylindrical pores) have 1D channels and inferior selectivity compared to CDC. The observed maximum adsorption difference of 1.33 mg/g between $^{18}O_2$ and $^{16}O_2$ under the present experimental setup during only 10 min of adsorption separation appears to be highly promising for dynamic separation. If we increase the sample amount to 1 kg of adsorbent, the available adsorption system can give rise to a preferential $^{18}O_2$ adsorption of 0.8 L(STP) and will be quite useful. A new type of adsorption-separation system with a large sample holder and mixed gas reservoir can be constructed that utilizes the initial large selective adsorption of $^{18}O_2$; this will enable human society to obtain a promising isotope separation technology, considering that the present $^{18}O_2$ separation technology using distillation requires 10 months.

**Breakthrough measurements**. Furthermore, we confirmed this by performing breakthrough measurements with an adsorption column packed with a comparatively large amount of commercially available ACFs to evaluate the separation performance. We used ACFs because the breakthrough experiments required larger sample sizes than the amount of the lab-synthesized CDC available. The experimental conditions and the selectivities for the breakthrough experiments are listed in Supplementary Table 2. A custom-made apparatus for measuring the breakthrough curves is schematically shown in Supplementary Fig. 18 (Supplementary Note 5). The breakthrough curves for ACF5 and

ACF10 are shown in Fig. 4a, b. The concentration $C/C_0$ of $^{16}O_2$ at the outlet in the initial stage is much larger than that of $^{18}O_2$ for both ACFs, clearly indicating the preferential adsorption of $^{18}O_2$ under continuous flow conditions using He as the carrier gas. The high $S$ values of 1.5 and 3.0 for ACF5 and ACF10, respectively, in the initial stage qualitatively correspond to the results from mixed gas adsorption experiments, as pictorially described in Fig. 4c. This reflects that the adsorbed molecules can mostly occupy the pore entrance sites during the initial stage of high selectivity and then diffuse to the inner pore region. The selectivity decreases with time as $^{18}O_2$ reaches full breakthrough and the flow approaches a stationary state. These selectivity values from the breakthrough measurement are much larger than the $S \sim 1.04$ of the current separation methods (Fig. 2f). However, the selectivity from the breakthrough experiment is smaller than that obtained from the mixed gas adsorption method discussed above.

**Theoretical selectivity of oxygen isotope adsorption**. Above, we have presented experimental results demonstrating $^{18}O_2/^{16}O_2$ selectivity but did not provide a clear explanation of the origin of these results. It is known that nuclear-quantum effects (NQEs) in the vibrational, rotational and translational motion of the condensed phase can each contribute to isotope selectivity[35–37]. $O_2$ molecules in the nanopores at low temperature form an ordered solid-like assembly in the nanopores due to the strong $O_2$–nanopore and $O_2$–$O_2$ interactions. We have performed simulations of $^{18}O_2$ and $^{16}O_2$ isotopes in MFI and SWCNTs using the path-integral formalism[35] to explicitly account for the NQEs in each of the degrees of freedom to identify the origin of the observed isotope selectivity. We note that isotopic selectivity is an inherently quantum mechanical phenomenon; we, therefore, employed path-integral molecular dynamics methods to compute the free energy differences for the gas and adsorbed phases of the different isotopes. In addition, adsorption selectivity is a thermodynamic equilibrium property, so that it is appropriate to use equilibrium path integral methods. In addition, we note that tunneling does not contribute in any way to the thermodynamic properties, and therefore does not need to be considered here. The experimental results give kinetic selectivity as a function of time. This quantity is not the true equilibrium selectivity but rather is related to a breakthrough process of dynamic adsorption. This process cannot be modeled using atomistic path integral methods because it would require detailed knowledge of the time-dependent pore filling process, including diffusion barriers through the irregularities in the adsorbent materials. Thus, it is only possible to model equilibrium selectivity. However, this modeling does provide a proof-of-concept that collective NQEs are responsible for the long-term (near-equilibrium) selectivity observed in the experiments. Details of the calculations are given in Supplementary Methods 1. We chose MFI and SWCNT because their atomistic structures are known precisely and therefore, these materials are amenable to accurate modeling. We note that these calculations represent equilibria and cannot account for the kinetic (time-dependent) selectivities observed in Fig. 3a, b but do explain the long-term (equilibrated) experimental results. We carried out simulations at both zero-loading (one $O_2$ per simulation cell) and at the loadings corresponding to the equilibrium amounts of $O_2$ adsorbed at a relative pressure $P/P_0$ of 0.12 at the temperatures of 90.2, 112, and 130 K to account for the effects of the adsorbate-adsorbate interactions on selectivity.

Path-integral calculations in the low $O_2$ coverage limit were carried out for SWCNTs having atom-center-to-atom-center tube diameters ranging from 0.54 to 1.35 nm (effective diameters of

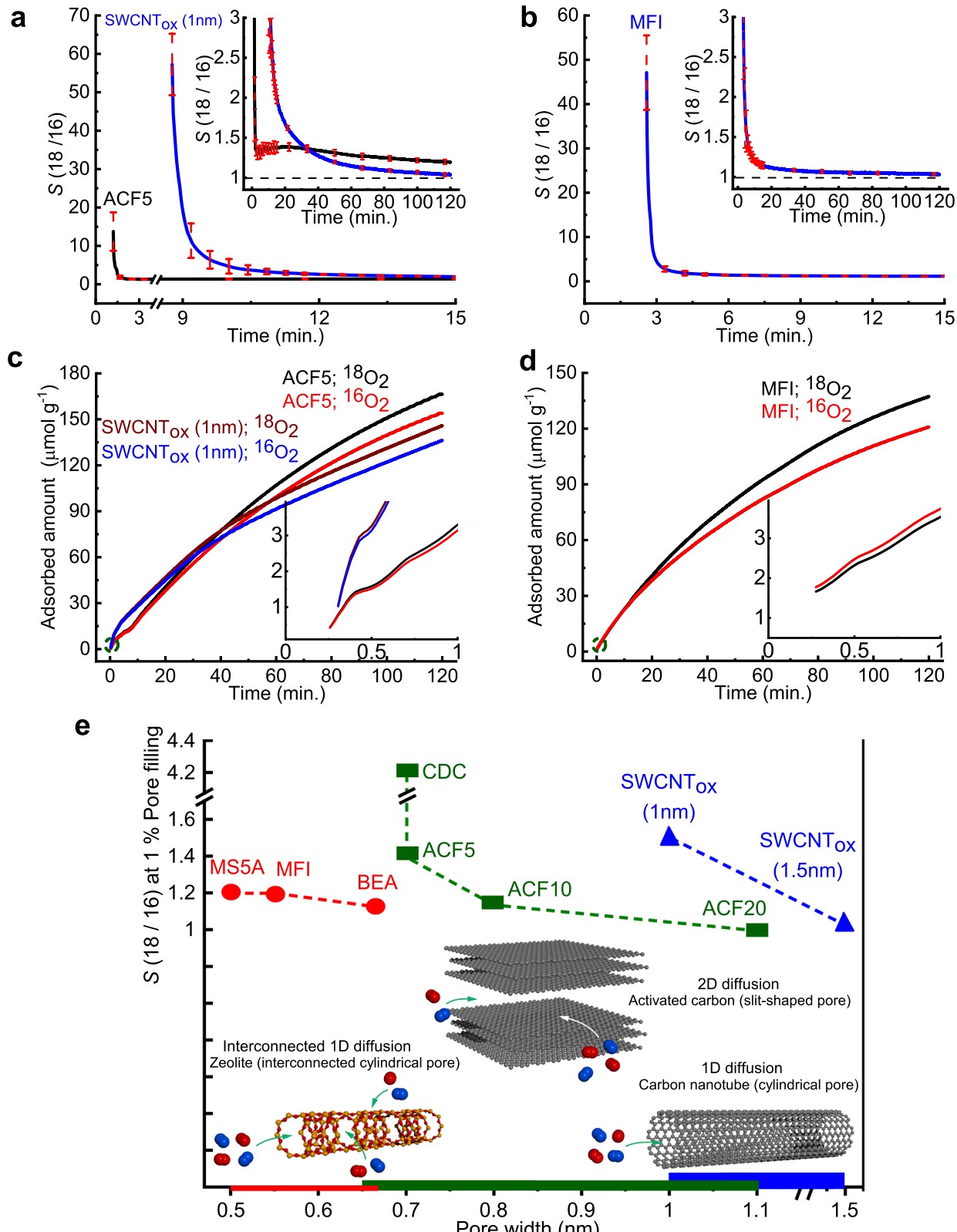

0.21–1.02 nm) to obtain the relationship between the selectivity and tube diameter; the results are shown in Fig. 5a and Supplementary Table 3. The (4,4) and (5,5) SWCNTs with extremely narrow diameters of 0.54 nm (0.21 nm effective diameter) and 0.69 nm (0.36 nm effective diameter), respectively, give high selectivity values at zero loadings through strong O2–SWCNT interactions, resulting in mass-dependent NQEs in the rotational, vibrational, and translational degrees of freedom. However, these pores are too small to induce the observable $O_2$ adsorption experimentally. In contrast, adsorption in the (7,7) and (10,10) SWCNTs at zero loading shows very low selectivities ($1.004 \pm 0.002$ and $1.006 \pm 0.0037$, respectively), indicating that

**Fig. 3 Pore geometry effect on oxygen isotope adsorption separation. a, b** Time evolution of $S$ for ACF5 and SWCNT$_{ox}$ (1 nm) and zeolite MFI (solid lines), respectively, during the initial few minutes at 112 K. Red dashed lines represent standard deviations derived from four measurements. Insets show $S$ for the full duration of the experiment. **c, d** Time evolution of the amount of adsorbed $^{18}O_2$ and $^{16}O_2$ on ACF5 and SWCNT$_{ox}$ (1 nm) and zeolite MFI. The adsorption amount is calculated from the difference of feed mixed gas and the unadsorbed mixed gas. Insets show the enlarged view of the circled initial region. **e** Pore geometry-dependent selectivity $S$ at 1% pore volume filling at 112 K. Slit pore (green rectangles) materials include carbide-derived carbon (CDC) and ACFs, cylindrical pore (blue triangles) materials are SWCNT$_{ox}$ (1 nm) and SWCNT$_{ox}$ (1.5 nm) and MS5A, MFI and BEA have channel cylindrical pores (red circle). An equimolar mixed gas ($^{18}O_2 + {}^{16}O_2$) is used in all experiments performed at 112 K. The accessibility of oxygen molecules for each pore is schematically shown at the bottom.

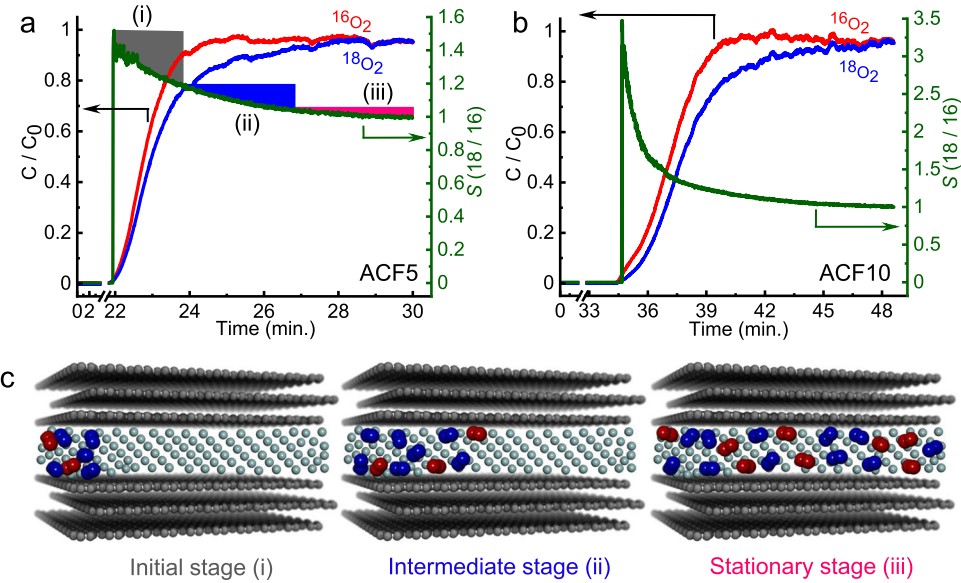

**Fig. 4 Selectivity for $^{18}O_2$ via breakthrough experiments. a** Breakthrough curves and selectivity on ACF5 at 103 K. **b** Breakthrough curves and selectivity on ACF10 at 103 K. The curves demonstrated the distinct separation of $^{18}O_2$ from $^{16}O_2$. Here, $C$ is the concentration of the component in the adsorption column outlet, and $C_0$ is the total concentration of all feed gases {$^{18}O_2(0.5) + {}^{16}O_2(0.5)$}. **c** Slit pore models demonstrate the selective adsorption process under He (grey spheres) flow corresponding to the high selectivity in the initial stage (i), intermediate (ii), and stationary (iii) stages of adsorption separation. The red and blue spheres represents adsorbed $^{16}O_2$ and $^{18}O_2$ molecules, respectively.

the experimentally observed results cannot be explained by simulations at low coverages.

We, therefore, simulated $O_2$ adsorbed in (7,8) and (10,10) SWCNTs with the actual diameters of 1.02 (0.7 nm effective diameter) and 1.36 nm (1.03 nm effective diameter), respectively, at 90.2 K using amounts of adsorbate close to that reported for SWCNT$_{ox}$(1 nm) at $P/P_0 = 0.12$ (Supplementary Fig. 19). The selectivities obtained from the finite loading simulation were $1.164 \pm 0.024$ for the (7,8) SWCNT and $1.02 \pm 0.007$ for the (10,10) SWCNT, similar to the long-time experimental value for the SWCNT$_{ox}$ (1 nm) of 1.09, as shown in the inset of Fig. 3a. The origin of the significant increase in selectivity with finite loading relative to the low coverage limit can be found in the highly ordered state of the $O_2$ molecules observed in the simulations. The initial random configuration shown on the left of inset Fig. 5a transforms to a highly ordered state (right) during the PIMD simulation. This highly ordered state gives rise to a collective NQE not obtained at low coverage, resulting in higher selectivity.

Furthermore, we calculated $O_2$ isotope selectivities for finite adsorption in MFI for $P/P_0 = 0.12$ to mimic experiments (Supplementary Figs. 20 and 21). The obtained selectivities were $1.092 \pm 0.028$, $1.095 \pm 0.019$, and $1.038 \pm 0.02$ for the temperatures of 90.2, 112, and 130 K, respectively (Fig. 5b and Supplementary Table 4), in agreement with the experimental results shown in Fig. 3b.

Our simulation results indicate that (1) narrow pores give higher selectivity and (2) the observed selectivity is a result of both pore confinement and the cooperative or collective NQEs

due to the ordering of the $O_2$ molecules when a high amount of $O_2$ molecules is adsorbed in the preferable pores (close to saturation). These observations are consistent with our hypothesis that narrow pore sites give rise to higher selectivities because they have higher confinement and are more likely to produce ordered $O_2$-adsorbed phases than the larger nanopores within CDC. Furthermore, we measured the difference in the activation energy ($E_a^{ads}$) for the overall adsorption rates of $^{18}O_2$ and $^{16}O_2$ on CDC at 112 K using the linear driving force model described in Supplementary Fig. 22 (Supplementary Methods 2). The $E_a^{ads}$ values obtained for $^{18}O_2$ and $^{16}O_2$ are $9.1 \pm 0.1$ kJ mol$^{-1}$ and $9.4 \pm 0.1$ kJ mol$^{-1}$, respectively. This small activation energy difference of $0.3 \pm 0.1$ kJ mol$^{-1}$ between $^{18}O_2$ and $^{16}O_2$ may be associated with the potential energy difference in the transition state for adsorption.

**Adsorption isotherms and energetics.** Figure 5c shows the experimental adsorption isotherms for $^{18}O_2$ and $^{16}O_2$ on CDC at 90.2, 112, and 130 K. All of the adsorption isotherms exhibit Type I behavior in the IUPAC classification, indicating the filling of oxygen molecules in highly uniform nanopores[38]. The $^{18}O_2$ uptake is essentially the same as the $^{16}O_2$ uptake below $P/P_0 = 10^{-4}$ (Fig. 5d). However, the $^{18}O_2$ uptake is slightly higher than that of $^{16}O_2$ at higher relative pressures (Supplementary Fig. 23). The isosteric heat of adsorption ($q_{st}$) was evaluated from the Clausius–Clapeyron equation using the adsorption isotherms at different temperatures (Supplementary Fig. 14 and

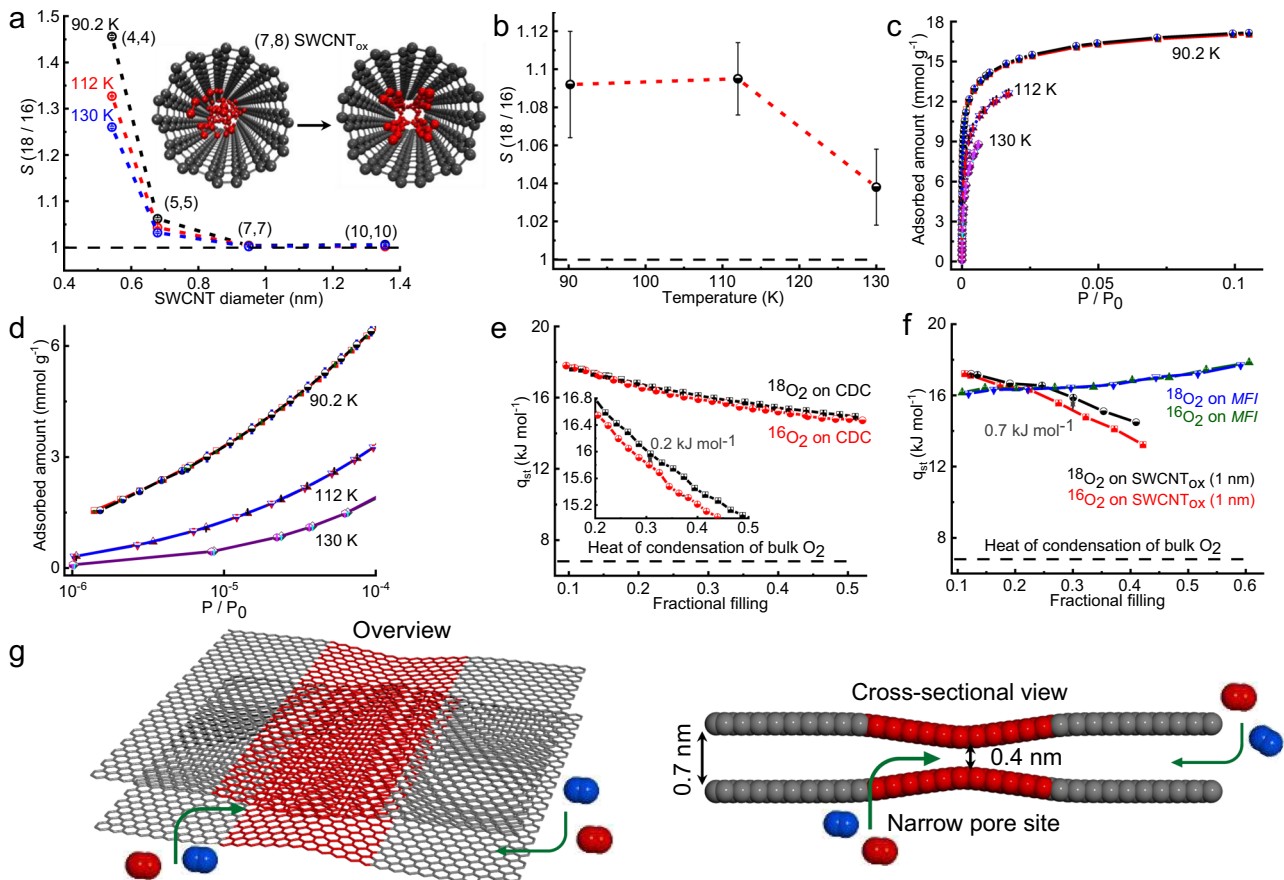

**Fig. 5 Theoretical selectivity and energetics of oxygen isotope adsorption. a** Dependence of the value of $S$ on the diameter of SWCNT$_{ox}$ and temperatures for the zero-loading conditions (one molecule of oxygen per cell). The temperatures considered were 90.2, 112, and 130 K. The inset shows a schematic illustration of the (7,8) SWCNT$_{ox}$ with adsorbed oxygen molecules. (Left) The initial configuration showing a random packing configuration of molecular oxygen. (Right) A transition to an ordered molecular packing of oxygen molecules after geometry optimization using force-field methods. **b** Selectivity $S$ for zeolite MFI with the temperature at $P/P_0 = 0.12$. Error bars are standard deviations calculated from four measurements. **c** $^{18}O_2$ and $^{16}O_2$ adsorption isotherms of carbide-derived carbon (CDC) at 90.2, 112, and 130 K. **d** Semilogarithmic plots of the isotherms in the low relative pressure region. **e** Isosteric heats of adsorption ($q_{st}$) of the $O_2$ isotopes on CDC as a function of fractional filling. Here, $^{16}O_2$ uptake is normalized to the maximum uptake at 90.2 K to obtain the fractional filling. **f** Isosteric heats of adsorption plotted versus fractional filling for SWCNT$_{ox}$ (1 nm) and MFI. Error bars are the standard deviations calculated fom three measurements. **g** Representative model showing the presence of narrow pore sites along with the 2D open entrances for oxygen isotopes ($^{16}O_2$ is red and $^{18}O_2$ is blue) in CDC.

Supplementary Methods 3). The $q_{st}$ values for both oxygen isotopes decrease from 18 kJ mol$^{-1}$ to 15 kJ mol$^{-1}$ at the filling of 0.5 (Fig. 5e), and is much larger than the heat of condensation of bulk $O_2$ ~6.8 kJ mol$^{-1}$ [39] due to the favorable interactions with the pores. The absolute adsorption amount difference of $^{18}O_2$ and $^{16}O_2$ is not sufficiently large to obtain an evident difference in the $q_{st}$ of the oxygen isotope molecules in the low fractional filling range. It is noteworthy that $q_{st}$ for $^{18}O_2$ is larger than that of $^{16}O_2$ above the fractional filling of 0.1, giving a difference of 0.2 ± 0. 005 kJ mol$^{-1}$ for CDC (inset in Fig. 5e). Similarly, the $q_{st}$ difference between $^{18}O_2$ and $^{16}O_2$ for SWCNT$_{ox}$ (1 nm) is 0.7 ± 0. 04 kJ mol$^{-1}$ at the filling of 0.3 (Fig. 5f) but much smaller differences are obtained at low coverage, consistent with our calculations. On the other hand, the $q_{st}$ of MFI gradually increases with fractional filling with no clear difference in the $q_{st}$ between $^{18}O_2$ and $^{16}O_2$ (Fig. 5f).

Since CDC gives the highest selectivity, we discuss the relationship between the pore shape and $q_{st}$ results. The large $q_{st}$ for CDC at the initial stage arises from the highly confined state of the oxygen molecules in the narrow pore sites with very deep interaction potential wells, forming the ordered molecular solids, as mentioned above. The $q_{st}$ gradually decreases with

fractional filling due to the changes to the adsorption sites of the shallower interaction potential wells. To evaluate the contribution of the narrowest pore sites toward the total adsorption, we simulated $N_2$ adsorption on our CDC model as shown in Fig. 2e left. The geometrical cumulative pore volume using nitrogen as a probe atom was calculated using Poreblazer[40]. It is noted that the pore highlighted within the small cyan circle contains spheres of diameter <0.4 nm and that space is not accessible to nitrogen (Supplementary Fig. 25). The cumulative pore volume of such narrow spaces is $V_{narrow-pore}$ ($d < 0.4$ nm) = 0.021 cm$^3$ g$^{-1}$, contributing to 3.5% of the total pore volume. These narrow sites exhibit 2D accessibility and this may facilitate the preferential adsorption of $^{18}O_2$ as pictorially demonstrated in Fig. 5g. The $^{18}O_2$ molecules strongly confined at the narrow spaces in the pores of CDC can form an ordered molecular solid-like assembly that leads to a collective NQE. Furthermore, 1D pores of SWCNTs and zeolites display less accessibility and weaker interaction potentials than the 2D narrow pores of CDC. Consequently, the collective NQE in SWCNTs and zeolites is not as remarkable as that in CDC. The interaction potential of ACFs is weaker than that of the narrow pore sites of CDC, even though ACF has well-accessible slit-shaped pores.

## Discussion

We have demonstrated very high $^{18}O_2$ adsorption selectivity of nanoporous materials that increases with decreasing temperature and has a strong dependence on the pore geometry of nanoporous materials. In particular, at approximately the boiling temperature of methane (112 K) CDC shows $S(^{18}O_2/^{16}O_2)$ exceeding $60 \pm 15$ for the initial stage of adsorption because $^{18}O_2$ molecules are kinetically and thermodynamically preferentially adsorbed over $^{16}O_2$. Such high adsorption selectivity was also evident for the separation of $^{13}CH_4$ with $S(^{13}C/^{12}C) = 56 \pm 6$. The highly efficient separation of $^{18}O_2$ or $^{13}CH_4$ evidenced in this study can be implemented in the industry by designing a rapid adsorption-separation instrument and will facilitate medical and other applications of these O and C isotopes. The CDC with highly accessible 0.4 nm wide pore spaces has a strong interaction potential for $O_2$ and gives the highest adsorption selectivity compared with activated carbon fibers, single-walled carbon nanotubes, and zeolites. Nanoporous materials consisting of graphitic walls and having an abundance of very narrow open pores are promising for achieving high separation selectivity and high separation capacity. The collective nuclear-quantum effect (NQE), which is different from the well-known translational motion-based QMS effect, can explain the equilibrium selective adsorptive separation of $^{18}O_2$ or $^{13}CH_4$ by subnanometer pores.

## Methods

**Synthesis and characterizations**. Pitch-based activated carbon fibers (ACF20, ACF10 and ACF5) were procured from Ad'all Co., Ltd. The CDC sample derived from TiC was synthesized by following the previously reported method[41,42]. Single-walled carbon nanotubes (SWCNT) grown by the CVD method were obtained from MEIJO NANO CARBON Co., Ltd. The samples were oxidized under dried air at 823 K for 10 min with the heating rate 1 K/minute. A typical steam-assisted crystallization synthetic method has been used for the synthesis of pure silica zeolite BEA[43]. Silicalite-1 or zeolite MFI was synthesized by following the seed surface crystallization mechanism discussed by Ren et al.[44]. Molecular sieves MS5A and MS4A were purchased from Nakarai Tesque. High purity $^{16}O_2$ (99.99995%) and $^{12}CH_4$ (99.9999%) were procured from JAPAN FINE PRODUCTS-JFP, $^{18}O_2$ (≥98%) from TAIYO NIPPON SANSO Corporation, and $^{13}CH_4$ (99%) from ISOTEC. The adsorption isotherms of the adsorbents were measured using a Microtrac MRB's apparatus (BELSORP MAX) after pre-evacuation at 423 K/3 h.

## Data availability

The authors declare that the data supporting the findings of this study are available from the authors and are presented in the paper and its supplementary information files.

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

## Acknowledgements

K.K. acknowledges project Japan Science Technology Agency (JST) CREST "Creation of Innovative Functional Materials with Advanced Properties by Hyper-nanospace Design" and partial support by the Grant-in-Aid for Scientific Research (B) (17H03039) and the OPERA JST project (JPMJOP1722). SKU was supported by the JST-CREST and OPERA projects. H.T., R.F., and F.V.-B. are supported by TAKAGI Co., Ltd. Funding for A.B. and J.K.J. was provided by the National Science Foundation under Award No. CBET1703266. Simulations were performed using the computational resources provided by the Center for Research Computing at the University of Pittsburgh and Extreme Science and Engineering Discovery Environment (XSEDE), which is supported by National Science Foundation grant number ACI-1548562, under allocation No. TG-DMR110091 and the Pawsey Supercomputing Centre, with funding from the Government of Australia and the Government of Western Australia. C.D.T. and I.S.M. are supported by the Australian Research Council under grant FT140100191. CDC synthesis was supported as part of the Fluid Interface Reactions, Structures and Transport (FIRST) Center, an Energy Frontier Research Center funded by the U.S. Department of Energy, Office of Science, and Office of Basic Energy Sciences.

## Author contributions

K.K. conceived the idea and supervised the whole project. S.K.U. being in the main charge of this research, designed the study, performed the experiments, and analyzed the experimental data. A.B., G.G., and J.K.J. designed and carried out the simulations using the path-integral formalism. Y.M. and A.K. performed breakthrough experiments. C.D.T. and I.S.M. generated atomistic models of CDCs via MD simulation. F.V.B. performed interaction potential calculation using the LJ and QFH potential. M.S. and M.M. performed the synthesis of the MFI and BEA zeolites. P.A., H.T., and R.F. discussed the results and helped in the analysis. S.K.U. and K.K. prepared the paper. A.B. explained the simulation results. S.K.U., P.A., J.K.L., Y.G., and K.K. edited the paper.

## Competing interests

The authors declare no competing interests.
