## [Peer Review File · Nature Communications]

REVIEWER COMMENTS

Reviewer #1 (Remarks to the Author):

This is an interesting manuscript, showing high kinetic selectivity for ^{18}O over ^{16}O in CDC due to nuclear quantum effects. The experiments are well described, and novel. In particular the authors have conducted experiments using a $^{18}\text{O}/^{16}\text{O}$ mixture, which is very important. Most works on quantum-kinetic effects on isotope separation, largely for D_2/H_2 separation have been based on pure component selectivities, and are not convincing. This is among the few that show high selectivity in mixtures.

Besides experiments the authors have performed simulations to determine adsorption selectivities; however, they are based on equilibrium and do not involve kinetic effects. This leads to the question of whether quantum tunneling is considered in their calculations, and how it influences selectivities. The authors may elaborate on this point.

Reviewer #2 (Remarks to the Author):

Manuscript on "Adsorption Separation of Heavier Isotope Gases in Subnanometer Carbon Pores"

The authors claim an extremely high selectivity for adsorption-based separation of $^{18}\text{O}_2$ from $^{16}\text{O}_2$ using nanoporous carbons. The manuscript contains an experimental and theoretical part, which does not explain or support the high observed selectivities.

The schematic setup in Figure S2 is not described nor explained properly. When is which valve opened, what is pump P2 doing etc.? The MS is connected via V7, which seems to be a needle valve. How can the MS give a quantitative result, if a more or less arbitrary gas amount will be extracted via the needle valve? The authors mention "the MS, the signal of which has been calibrated in advance.", however, no calibration procedure is given.

Apparently, this setup is used to determine the results, e.g., in Figure 1, S6 and S7. Figure S6 shows the selectivities versus time, but what is happening, e.g., for the 50% mixture, prior to 40 min? The adsorption amount is shown in Figure S7 starting from time 0 min for each isotope. How is the adsorbed amount of different isotopes measured for a mixture, if only the gas flow behind the sample cell will be analyzed by MS? How can one determine 2 micro-mol/g on a 50 mg sample

(Figure S7 inset), i.e., detect 0.1 micro-mol gas amount accurately? Why starts the adsorption immediately, while the selectivity above 40 min? There is a clear inconsistency between Figures S6 and S7.

The same arguments hold for Figure 2. How can one measure the amounts of $^{18}\text{O}_2$ and $^{16}\text{O}_2$ separately in a mixture? Does a pore volume filling of 0.6% make any sense? This is near zero coverage of the surface.

Equation (1) defines the adsorption selectivity, but how is the mole fraction in the adsorbed phase determined and how in the gas phase? This is never explained.

The results are highly questionable, since no explanation is given about the time immediately after exposure to the mixture. For all samples and isotope mixtures the selectivity versus time curves are very similar, i.e., starting at a high value of 50 to 80 and then rapidly dropping to nearly $S=1$. These similarities for different materials and gases may hint at an experimental artifact rather than a real separation effect. In total, the adsorbed amount of gas is in the range of tens of micro-mol, which is an almost negligible adsorption amount compared to the specific surface areas given in Table S1. Furthermore, these extremely low adsorption values are of no use for any technological application and for higher values the selectivity is always close to one.

The theoretical calculations yield very low selectivities Table S3 and S4, i.e. near 1 except for the narrow (4,4) SWCNT. Table S3 does not say which selectivity, is it the same as in Table S4? These very low selectivities are in line with the small difference in isosteric heats determined experimentally. However, cannot explain or reproduce the high values obtained by the experiments at ultra-low adsorption amounts.

Reviewer #3 (Remarks to the Author):

The manuscript by S. Ujjain et al. reports on a method to separate isotope gases such as methane and oxygen using dynamic separation in nanoporous structures at low temperatures. The authors propose a novel method where the separation is performed at low temperature and in various nanoporous structures. The authors identify more efficient separation in some structures compared to others, and changing separation features with time. They also perform numerical simulations to identify the origin of these different separation features.

I find the experiments to be well detailed and impressive work. We rely on isotope separation for a number of applications and the methods available today for separation are very energy consuming — therefore I find the topic of the paper very relevant.

However, I find the manuscript fails to answer convincingly to either one of the 2 scientific questions at hand here (1) how is this method really comparing to other methods in terms of energetics, cost etc and is it worth it ? (2) what is the critical effect responsible for enhanced selectivity ?

(1) Although comparison to other methods is stated in terms of operating selectivity S , it is unclear how the energetic efficiency compares. Their system works at low temperature and therefore requires cooling, which is energy consuming. Their system also ages and needs to be cleaned. How expensive is it to form these materials, especially at large scales ?

(2) The entire discussion on enhanced selectivity is very obscure, in part because the authors drown the reader in an extensive list of results, failing to draw from that a concise and clear picture. Experimentally, the dependence on the pore width is not necessarily always going down. The porous arrangement/geometry changes a lot the selectivity. This is not discussed at all. Further, authors seem to believe that very narrow regions are critical for selectivity, but show no evidence of that — Fig. 4i is an illustration, not a proof. Also... a very enigmatic claim of nuclear quantum effects playing a role is brought forward, with no evidence, in particular no benchmark simulation to compare to and no details. Finally, the reason for very different time scales is not at all discussed.

To wrap it up, I find the manuscript in its current state to be simply not conclusive enough for me to recommend publication in Nature Communications.

A few specific comments

The authors could discuss the applicability to other isotope separation

Line 78 does not make sense, and I do not believe Ref [23] concurs on the term “quantum sieving” though it hints to possible quantum effects in separation processes, which is not entirely the same thing.

Line 117 “the pore widths”.. what calculations are you talking about ? It seems that your chemical structures are first equilibrated so I don't see why you would need that. Why do you subtract that value ?

I think an explanation of why S decreases should be brought forward early on for the very broad audience.

Fig S10 and S17 have layout issues.

Fig 3c is a real simulation ? Or is a picture view ?

The discussion from line 338 lacks clarity and proofs.

**Title: Adsorption Separation of Heavier Isotope Gases in
Subnanometer Carbon Pores**

Sanjeev Kumar Ujjain, Abhishek Bagusetty, Yuki Matsuda, Hideki Tanaka, Preety Ahuja,
Carla de Tomas, Motomu Sakai, Fernando Vallejos-Burgos, Ryusuke Futamura, Irene
Suarez-Martinez, Masahiko Matsukata, Akio Kodama, Giovanni Garberoglio, Yury
Gogotsi, J. Karl Johnson, Katsumi Kaneko

Manuscript Ref. No. NCOMMS-20-20283-T

Response to Reviewer's comments:

At first, we appreciate your consideration of our manuscript (NCOMMS-20-20283-T). We appreciate Editor and Reviewers for their kind comments, suggestions and precious time. We have considered their suggestions carefully to make changes in the manuscript so as to improve our results and figures (Highlighted in green color).

A detailed point-by-point response to the editor and reviewers' comments is attached below:

Reviewer #1 (Remarks to the Author):

This is an interesting manuscript, showing high kinetic selectivity for ^{18}O over ^{16}O in CDC due to nuclear quantum effects. The experiments are well described, and novel. In particular, the authors have conducted experiments using a $^{18}\text{O}/^{16}\text{O}$ mixture, which is very important. Most works on quantum-kinetic effects on isotope separation, largely for D_2/H_2 separation have been based on pure component selectivities, and are not convincing. This is among the few that show high selectivity in mixtures.

Comment 1.1# Besides experiments the authors have performed simulations to determine adsorption selectivities; however, they are based on equilibrium and do not involve kinetic effects. This leads to the question of whether quantum tunneling is considered in their calculations, and how it influences selectivities. The authors may elaborate on this point.

Response 1.1# The calculations performed do not include quantum tunneling because

there is no barrier in the process being modeled. Moreover, it is not clear how quantum tunneling would apply to a dynamic adsorption process, since there is no well-defined barrier associated with this process. Therefore, including quantum tunneling is not appropriate for this problem.

Reviewer #2 (Remarks to the Author):

Manuscript on "Adsorption Separation of Heavier Isotope Gases in Subnanometer Carbon Pores"

Comment 2.1# The authors claim an extremely high selectivity for adsorption-based separation of $^{18}\text{O}_2$ from $^{16}\text{O}_2$ using nanoporous carbons. The manuscript contains an experimental and theoretical part, which does not explain or support the high observed selectivities.

Response 2.1# The theoretical modeling provides a proof-of-concept that indeed quantum effects lead to selectivity at equilibrium and the results from modeling agree quantitatively with the experimental results at long times. The extremely high selectivities are due to kinetic effects, which cannot at present be accurately modeled with path integral methods. It is really essential that the equilibrium theory can predict the selectivity of more than 1.15 for the (7,8) SWCNT and 1.09 for MFI zeolite, because even these selectivities are much larger than the observed values in the literatures. The proposed theory gives the firm basis on the equilibrium selectivity for oxygen isotope separation observed in this study. Then, we need to establish new quantum kinetic theory for the observed kinetic selectivity in the next stage with a worldwide collaboration.

Comment 2.2# The schematic setup in Figure S2 is not described nor explained properly. When is which valve opened, what is pump P2 doing etc.? The MS is connected via V7, which seems to be a needle valve. How can the MS give a quantitative result, if a more or less arbitrary gas amount will be extracted via the needle valve? The authors mention “the MS, the signal of which has been calibrated in advance.”, however, no calibration procedure is given.

Response 2.2# The working of the flow-type dynamic adsorption separation instrument is described in the Revised Supplementary material. The role of different valves, pump,

needle valve and MS calibration has been described in detail.

Comment 2.3# Apparently, this setup is used to determine the results, e.g., in Figure 1, S6 and S7. Figure S6 shows the selectivities versus time, but what is happening, e.g., for the 50% mixture, prior to 40 min? Why starts the adsorption immediately, while the selectivity above 40 min? There is a clear inconsistency between Figures S6 and S7. The adsorption amount is shown in Figure S7 starting from time 0 min for each isotope.

Response 2.3# The MS perform on-line analysis of the unadsorbed gaseous phase of mixed gas. When mixed gas is introduced to the adsorption cell maintained at 112K, almost all isotope gas ($^{18}\text{O}_2$) introduced is adsorbed by CDC of large pore volume (0.54 ml g^{-1}), while only few $^{16}\text{O}_2$ molecules which remain unadsorbed can be detected by MS. This can be reflected from the difference in adsorption amount of $^{18}\text{O}_2$ and $^{16}\text{O}_2$ even in the initial adsorption time. The adsorption selectivity S for $^{18}\text{O}_2$ against $^{16}\text{O}_2$ is defined as:

$$S(^{18}\text{O}_2/^{16}\text{O}_2)_{(\text{ads-g})} = \frac{(^{18}\text{O}_2/^{16}\text{O}_2)_{\text{ads}}}{(^{18}\text{O}_2/^{16}\text{O}_2)_{\text{g}}}$$

As the dividing factor $(^{18}\text{O}_2/^{16}\text{O}_2)_{\text{g}}$ is zero, the whole expression becomes infinite. Consequently, the selectivity determination is possible only when a few $^{18}\text{O}_2$ molecules remain unadsorbed which can be analyzed by the MS. That is why, even though we can determine the adsorption amount from 0 min. of the adsorption experiment, but selectivity can be analyzed only after 40 min., when pore volumes of CDC is sufficiently filled to allow a few $^{18}\text{O}_2$ molecules to reach MS. The selective adsorption capacity plays extremely significant role in the time period after which selectivity analysis can be performed. This can be well explained using the figure below.

Figure L1. Time courses of mass intensity at room temperature and 112 K with or without adsorbent.

The **Figure L1 a** shows time course of mass intensity in two regions: Without adsorbent at room temperature (in blue dashed box) and through adsorbent maintained at 112 K (in green dashed box). Figures L1 **b**, **c** and **d** show their magnified view. The molar fraction of mixed isotope gas as determined from the no adsorbent region is, $n(^{18}\text{O}_2 : ^{16}\text{O}_2) = 1 : 1$ (Figure **b**). When the mixed isotope gas is passed through adsorbent maintained at 112 K (Figure **c** & **d**), almost all isotope gas $^{18}\text{O}_2$ is adsorbed for initial 9.7 minutes and the $^{16}\text{O}_2$ which remain unadsorbed can only be detected by the MS as can be seen from Figure **d**. Consequently, the selectivity determination before 9.7 minutes or immediately after exposure of the adsorbent to the mixture in present experiment is not possible. The selectivity can only be determined after 9.7 minutes, when MS starts detecting the $^{18}\text{O}_2$ gas molecules also. At 10 minute, it can be observed that the difference in the mass intensity of $^{18}\text{O}_2$ and $^{16}\text{O}_2$ is very large, giving the high selectivity. The above experiment is performed using 300 Pa of mixed oxygen isotope gas introduced at a dosing rate $\sim 6 \text{ ml min}^{-1}$. So, it can be inferred that the initiation of selectivity time is dependent on the amount, dosing rate and molar ratio of feed mixed gas as well as the adsorption temperature.

This is also reflected in the initial time of selectivities, which show dependence on dosing rate (see **Figure L2**), mixed dosing gas molar ratio (Fig. 1b) and temperature (Fig. 1d). **Figure L2** shows the dependence of the time-change of the selectivity on the dosing rate from 1 ml min^{-1} to 1.4 ml min^{-1} of the mixture gas. It can be observed that with increasing dosing rate, the initial selectivity time decreases. This implies, the time period after which a few of $^{18}\text{O}_2$ molecules remain unadsorbed decreases with increasing dosing rate. Similar tendency can also be seen from molar ratio of mixed dosing gas dependence of selectivity graph shown in Fig. 1b. With increasing $^{18}\text{O}_2$ concentrations, the initial time decrease. The temperature-dependent selectivity graph shown in Fig. 1d, reaffirms that on increasing temperature, the adsorption capacity decreases and hence the $^{18}\text{O}_2$ molecules become available for MS detection in early time and hence selectivity can be observed earlier. Thus, this dynamic mixed gas method is quite useful to detect the selective adsorption for various samples.

The present explanation is included in the revised manuscript Supplementary material Fig. S7 and S8.

Figure L2. *Dependence of the time-change of the selectivity on the dosing rate*

Comment 2.4# How is the adsorbed amount of different isotopes measured for a mixture, if only the gas flow behind the sample cell will be analyzed by MS?

Response 2.4# *The MS analyzes the concentration of the component gas in the mixed gas of both downstream (i.e. mixture gas before introduction to the sample cell) and upstream (mixture gas remains unadsorbed during the dynamic adsorption separation). Thus we determine the mole fraction of the component gas in the adsorbed mixed gas.*

Comment 2.5# How can one determine 2 micro-mol/g on a 50 mg sample (Figure S7 inset), i.e., detect 0.1 micro-mol gas amount accurately?

Response 2.5# *Number of mole of gas can be evaluated using Gas laws, Laws that relate the pressure, volume, and temperature of a gas. Quadrupole Mass Spectrometers M-201QA-TDM has a high-sensitivity, the minimum detected partial pressure limit is 1.0×10^{-12} Pa or less. The adsorption amount discussed here is determined from the mass intensity under different partial pressure observed. The relationship between the mass intensity and partial pressure of individual gases is calibrated in advance, as discussed in the Revised Supplementary material under Fig. S3.*

Comment 2.6# The same arguments hold for Figure 2. How can one measure the amounts of $^{18}\text{O}_2$ and $^{16}\text{O}_2$ separately in a mixture?

Response 2.6# *It can be done using the relationship between linearity of “partial pressure” and “mass intensity” via calibration approaches. Direct calibration for*

overlapping signals is given by:

$$I_i = K \sum a_{ij} P_j$$

Where, I_i - ion current of the mass M

K – instrument constant, related to the setup settings

a_{ij} – calibration factor determined from the slope of partial pressure vs intensity

P_j – partial pressure of the j -th component

For two component system, the linear equations could be written as follows:

$$\begin{bmatrix} I_1 \\ I_2 \end{bmatrix} = \begin{bmatrix} a_{11} & a_{21} \\ a_{12} & a_{22} \end{bmatrix} \cdot \begin{bmatrix} P_1 \\ P_2 \end{bmatrix}$$

i.e.
$$\begin{bmatrix} I_{36} \\ I_{32} \end{bmatrix} = \begin{bmatrix} a_{1802(36)} & a_{1602(36)} \\ a_{1802(32)} & a_{1602(32)} \end{bmatrix} \cdot \begin{bmatrix} P_{36} \\ P_{32} \end{bmatrix}$$

Such linear equations can be solved considering the following example:

$$X = ak + bq$$

$$Y = ar + bs$$

then,

$$\begin{bmatrix} k & q \\ r & s \end{bmatrix} \cdot \begin{bmatrix} a \\ b \end{bmatrix} = \begin{bmatrix} X \\ Y \end{bmatrix}$$

[Calibration factor] . [Unknown concentration] = [Experimental ion currents]

i.e.
$$\begin{bmatrix} a \\ b \end{bmatrix} = \begin{bmatrix} k & q \\ r & s \end{bmatrix}^{-1} \cdot \begin{bmatrix} X \\ Y \end{bmatrix}$$

this implies
$$a = \frac{sx - qy}{ks - qr} \quad b = \frac{ky - rx}{ks - qr} \quad b/a = \frac{ky - rx}{sx - qy}$$

This is how the amount or mole fraction of $^{18}\text{O}_2$ and $^{16}\text{O}_2$ can be determined separately from a mixture in adsorbed phase and gas phase. These explanations are included in the Revised Supplementary material Fig. S3.

Comment 2.7# Does a pore volume filling of 0.6% make any sense? This is near zero coverage of the surface.

Response 2.7# Pore volume filling of the adsorbents under present study depends on various factors including amount of mixture gas introduced, dosing rate of introduction,

temperature etc. If adsorption experiment is performed with higher amount of mixed gas at the higher flow rate, the pore volume filling % becomes larger, as demonstrated in **Figure L3**. When we increase the mixed gas pressure up to 240 Pa with the dosing rate of 3.1 ml min^{-1} , the pore volume filling is about 5 %, being very realistic. Thus, we have obtained the promising results for oxygen isotope separation in this study. However, the present kinetic mixed gas adsorption system has a limitation for increasing the sample amount and gas volume supplied, because we must do a careful experiment using the mixed gas at a constant temperature with an appropriate sample holder and sample gas holder. The above 5 % of the pore volume filling is the maximum value from the present adsorption system. If we construct a larger sample holder and the mixed gas reservoir with a large capacity cooling system, separation corresponding to the pore filling larger than 10 % is possible. However, the main purpose of the present work is to show the distinct selectivity of nanoporous materials, in particular, CDC for oxygen isotopes near 112 K. The 0.6 % filling experiment of high sensitivity is valuable for scientific discussion on the role of pore structures of different class of adsorbents for separation of oxygen isotopes and qualitative selection of the optimum adsorbent which can demonstrate high selectivity for oxygen or carbon isotope separation which is currently quite difficult because of the very low selectivity. Such fundamental studies can develop new isotope separation engineering in future. Revised Supplementary material Fig. S12.

Figure L3. Time courses of the pore volume filling on introduction of the mixed gas of 240 Pa.

Comment 2.8# Equation (1) defines the adsorption selectivity, but how is the mole fraction in the adsorbed phase determined and how in the gas phase? This is never explained.

Response 2.8# The above Response 2.6# has explained the procedure to determine the mole fraction in the adsorbed phase and the gas phase. Explanations are included in revised manuscript Supplementary material Fig. S3.

Comment 2.9# The results are highly questionable, since no explanation is given about the time immediately after exposure to the mixture.

Response 2.9# It is related to the large pore volume of CDC (0.54 ml g^{-1}) which can adsorb almost all mixed gas introduced to the adsorption cell. Although, during initial time all $^{18}\text{O}_2$ molecules are adsorbed and few $^{16}\text{O}_2$ remain unadsorbed, but we can observe the selectivity only from the time at which few $^{18}\text{O}_2$ remain unadsorbed. It is explained in Response 2.3# and also included in the revised manuscript. Also the related reason is given in Response 2.7#; our experimental system is designed for accurate determination of the kinetic adsorption at low temperature using a small sample holder and the mixed gas reservoir.

Comment 2.10# For all samples and isotope mixtures the selectivity versus time curves are very similar, i.e., starting at a high value of 50 to 80 and then rapidly dropping to nearly $S=1$. These similarities for different materials and gases may hint at an experimental artifact rather than a real separation effect.

Response 2.10# Here we will show two evidences on our experimental results.

(1) The selectivity vs. time depends on the pore structure.

The similarity comes from the presence of appropriate pore structures which induces the high selective adsorption of $^{18}\text{O}_2$ near 112 K, which is explained in the Response 2.3#. If the pore structure is not fit for the selective $^{18}\text{O}_2$ adsorption, the selectivity vs. time curves are completely different from those shown in Fig.1d and Fig.2a and b. We have the essentially important data that carbons having inaccessible pores for oxygen molecules do not show any selective adsorption. The following Figure L4 shows the time course of the selectivity for carbon molecular sieves (CMS) having pore width of $\sim 0.3 \text{ nm}$ and zeolite MS4A having pore width $\sim 0.4 \text{ nm}$. The observed selectivity is 1 for

both samples; strictly speaking the selectivity of MS4A is slightly larger than 1 in the initial stage. As the molecular size of an oxygen molecule in the range of 0.3 to 0.4 nm, oxygen molecules cannot be adsorbed in those pores at 112 K; oxygen molecules having the quadruple moment induce serious pore blocking effect even for the pores of MS4A. Then, the initial selectivity for MS4A is slightly larger than 1; the pore blocking prohibits the selective adsorption of $^{18}\text{O}_2$ and then the selectivity becomes 1 after 30 min. $^{18}\text{O}_2$ is not preferentially adsorbed in the pores of CMS and then the selectivity is 1.0. Thus, optimum pores are necessary for giving the selective $^{18}\text{O}_2$ adsorption. Another example is the time change of ACF20 having a larger pore width of 1.1 nm. As ACF has a sharp pore size distribution, we can examine the accurate effect of the pore size on the selective adsorption as shown in Fig. 2 e. As pores of ACF20 are too large to induce the selective $^{18}\text{O}_2$ adsorption, the selectivity increases gradually with time in the initial stage and reaching the broad maximum, as shown in the Fig. S15a and **Figure L4**. Oxygen molecules are strongly adsorbed on the pore walls of ACF20 in the initial stage without showing the selectivity. The effective pore width of the residual slit space between both oxygen monolayers on the pore wall is about 0.4 nm, which exhibits the preferential adsorption of $^{18}\text{O}_2$. This is the reason for giving the low maximum in the observed selectivity.

Figure L4. The selectivity vs. time curves for carbon molecular sieve (CMS-0.3 nm), zeolite MS4A-0.4 nm, and ACF20(1.1 nm) at 112 K.

(2) We can observe the preferential $^{18}\text{O}_2$ adsorption larger than $^{16}\text{O}_2$ by 0.037 mmol/g (0.8 mL(STP)/g), which is promising for future application. The decrease of the adsorption amount ratio of $n(^{18}\text{O}_2) / n(^{16}\text{O}_2)$ with time is evidenced by desorption of adsorbed oxygen molecules.

Adsorption separation experiments are performed using CDC adsorbent for only 10 min, 15 min and 20 min at 112 K. The molar fraction of feed and adsorbed mixture isotope gas is monitored using time course of mass intensity. The molar fraction of feed gas is kept constant $n(^{18}\text{O}_2 / ^{16}\text{O}_2) = 1$. We determined the molar adsorption amounts for initial 10 min, 10-15 min and 15-20 min by desorption. Here, almost all $^{18}\text{O}_2$ and $^{16}\text{O}_2$ are adsorbed during the initial 5 min, as mentioned in **Response #2.3** and then we measured the adsorption amount difference for initial 10 min. However, the molar adsorption amount of $^{18}\text{O}_2$ is much larger than that of $^{16}\text{O}_2$ for 5-10 min, giving the considerably large difference as shown in **Figure L5**. **Figure L5** shows the time course of the molar adsorption amount ratio $n(^{18}\text{O}_2 / ^{16}\text{O}_2)$. Here, the abscissa "time" in **Figure L5** indicates time after which adsorption was stopped. This figure explicitly indicates that $n(^{18}\text{O}_2 / ^{16}\text{O}_2)$ decreases with time, being qualitatively similar to the data shown in Fig. 1b, d and h, although the absolute values in Fig. 1b, d and h are much larger than these values due to the reason described in **Response #2.3**.

The observed molar fraction of desorbed mixed gas is $n(^{18}\text{O}_2 / ^{16}\text{O}_2) = 1.4 \pm 0.005$ after adsorption for 10 min. While the desorbed mixed gas has $n(^{18}\text{O}_2 / ^{16}\text{O}_2) = 1.3 \pm 0.006$ after 20 min. This shows that $\sim 8\%$ higher adsorption of $^{18}\text{O}_2$ takes place during 10 min, which corresponds to a maximum difference of **1.33 mg/g** between $^{18}\text{O}_2$ and $^{16}\text{O}_2$. Then, if we increase the sample amount to 1 kg-adsorbent, the available adsorption system can give rise to a **preferential $^{18}\text{O}_2$ adsorption of 0.8 L(STP)**, being quite useful. If we can construct a new type of adsorption separation system which utilizes the initial large selective adsorption of $^{18}\text{O}_2$, human society can get a promising isotope separation technology, taking into account that the present $^{18}\text{O}_2$ separation technology using distillation takes 10 months.

The above results are included in the revised manuscripts and Fig. S14 and Fig. S16.

Figure L5. *Sampling time dependence of molar adsorption amount ratio n ($^{18}\text{O}_2/^{16}\text{O}_2$)*

Comment 2.11# In total, the adsorbed amount of gas is in the range of tens of micro-mol, which is an almost negligible adsorption amount compared to the specific surface areas given in Table S1. Furthermore, these extremely low adsorption values are of no use for any technological application and for higher values the selectivity is always close to one.

Response 2.11# *The present separation of $^{18}\text{O}_2$ and $^{16}\text{O}_2$ has been carried out with energy-intensive technology. We must find a new scientific principle which can be applicable to a new technology. In this regard, we have studied this adsorption-mediated separation with nanoporous materials using a small size experimental system which is quite sensitive to selective adsorption. Also, we need to show the wide applicability of this new method to other isotope gases such as $^{13}\text{CH}_4$ and $^{12}\text{CH}_4$. The conventional quantum molecular sieving has been widely studied from a scientific view of point, showing the effective separation for hydrogen or helium isotopes. However, it cannot be applied to heavier isotope gases such as oxygen and methane. Our finding can open a new door for an efficient separation of heavier isotope gases, even though our separation amount is not so large at this stage.*

The main object of this study is to demonstrate that oxygen or methane isotopes or other heavier isotopes compared to hydrogen or helium can be separated via adsorption method. Then, we developed a new mixed gas adsorption equipment which can detect the selectivity with high sensitivity. However, the absolute adsorption amount is in the range of 150-180 $\mu\text{mol g}^{-1}$. If we construct a new adsorption system having a larger sample holder and mixed gas reservoir, much larger adsorption amounts are available.

Even the present adsorption system can increase the absolute adsorption amount to $\sim 1 \text{ mmol g}^{-1}$, as follows.

*When we increase the amount of feed mixed isotope gas or the dosing rate, the adsorption amounts show increment as shown in **Figure L3**. **Figure L3** shows the changes of adsorption amount with time under the pressure = 240 Pa at the flow rate =*

1.5 ml min^{-1} and 3.1 ml min^{-1} . The corresponding adsorption amounts are $\sim 0.4 \text{ mmol g}^{-1}$ (9 ml(STP)/g) and $\sim 0.9 \text{ mmol g}^{-1}$ (20 ml(STP)/g) with maintaining similar selectivity trend. The CDC adsorbent in the present study can attain a maximum adsorption amount of $\sim 13 \text{ mmol g}^{-1}$ (290 ml(STP)/g) at $P/P_0 = 0.02$ as shown in Fig. 4d. The extremely high cost of $^{18}\text{O}_2$ isotope gas limits us from doing adsorption experiments with very high amount of feed gas. Moreover, the experiments performed in **Response 2.10#** shows that the adsorption amount attains a difference of 1.33 mg/g between $^{18}\text{O}_2$ and $^{16}\text{O}_2$ during initial 10 min. of adsorption. This value is considerably high and promising for dynamic separation.

Figure L3. Time courses of the pore volume filling on introduction of the mixed gas of 240 Pa.

Comment 2.12# The theoretical calculations yield very low selectivities Table S3 and S4, i.e. near 1 except for the narrow (4,4) SWCNT. Table S3 does not say which selectivity, is it the same as in Table S4? These very low selectivities are in line with the small difference in isosteric heats determined experimentally. However, cannot explain or reproduce the high values obtained by the experiments at ultra-low adsorption amounts.

Response 2.12# The selectivity is defined in the same way for Tables S3 and S4, as given by Eq. (S2). The values in Table S4 are for experimentally realistic loading of the pore. These values are significantly different from 1, because the current technology uses the selectivity ≈ 1.04 . The calculated values are in line with the experimental values at long time in Fig. 2b. The calculations prove that the experimentally observed

selectivity at long times (equilibrium) is due to quantum effects.

Reviewer #3 (Remarks to the Author):

The manuscript by S. Ujjain et al. reports on a method to separate isotope gases such as methane and oxygen using dynamic separation in nanoporous structures at low temperatures. The authors propose a novel method where the separation is performed at low temperature and in various nanoporous structures. The authors identify more efficient separation in some structures compared to others, and changing separation features with time. They also perform numerical simulations to identify the origin of these different separation features. I find the experiments to be well detailed and impressive work. We rely on isotope separation for a number of applications and the methods available today for separation are very energy consuming, therefore I find the topic of the paper very relevant.

However, I find the manuscript fails to answer convincingly to either one of the 2 scientific questions at hand here (1) how is this method really comparing to other methods in terms of energetics, cost etc and is it worth it ? (2) what is the critical effect responsible for enhanced selectivity?

Comment 3.1# Although comparison to other methods is stated in terms of operating selectivity S , it is unclear how the energetic efficiency compares. Their system works at low temperature and therefore requires cooling, which is energy consuming. Their system also ages and needs to be cleaned. How expensive is it to form these materials, especially at large scales?

Response 3.1#

We agree with the reviewer's comment that our system works at low temperature and therefore requires cooling, which is energy consuming. However, here we have proposed to utilize the cryogenic energy of liquid methane (boiling temperature 112 K) to attain the adsorption temperature. We are proposing the collaboration with natural gas technology which must utilize their cryogenic energy efficiently. Distillation technology needs sophisticated experimental set-up with a large distillation tower standing about 30 meters high. Furthermore, it takes about 10 months alongwith keeping low temperature. The selectivity used in the present technology is 1.04. If we use the selectivity of 2, the time becomes much shorter (for example, at least 10 times shorter), saving the cooling energy markedly. At this moment, price of porous carbon is 10 USD/kg, but we must develop high quality porous carbon as suggested in this paper.

The price of the new porous carbon may be 30 USD/kg, depending on the production amount. If new method will be widely used, the price should be close to 10 USD/kg. We did not estimate the necessary energy of this proposed method yet. In the next stage, a chemical engineering estimation must be done under the assumption of nanoporous carbon having optimum pore structure proposed in this study.

Comment 3.2# The entire discussion on enhanced selectivity is very obscure, in part because the authors drown the reader in an extensive list of results, failing to draw from that a concise and clear picture. Experimentally, the dependence on the pore width is not necessarily always going down. The porous arrangement/geometry changes a lot the selectivity. This is not discussed at all.

***Response 3.2#** The present study is conducted to demonstrate that oxygen or methane isotopes or other heavier isotope gases compared to hydrogen or helium can be separated via adsorption method. However, their adsorption mechanism is different from conventional Quantum molecular sieving separation. In order to confirm this, a series of adsorbents having different pore geometry have been presented in the current study. Reviewer's comment that the porous arrangement/geometry changes a lot the selectivity is in concurrence with the results presented in Fig. 2e. The results on the effect of porous arrangement/geometry changes is presented in Fig. 2 and Supplementary Fig. S15 and S16. Furthermore, the relationship between the observed selectivity, pore shape and isosteric heat of adsorption (q_{st}) results are discussed on page 22-23 of the manuscript. More discussions are included in revised manuscript while describing Fig. 2e*

Comment 3.3# Further, authors seem to believe that very narrow regions are critical for selectivity, but show no evidence of that Fig. 4i is an illustration, not a proof.

***Response 3.3#** We understand that conceptualizing the kinetics issue from theory for narrow regions are critical. However, I do not think it is practical to try to use the illustrative model shown in Fig. 4i to observed time dependent selectivity. The calculations would still be equilibrium in nature. There is, to the best of our knowledge, no way to include kinetic rates in these very complicated but rigorous calculations. What the modeling has clearly shown is that the long-time selectivities from experiments are consistent with the model. In our opinion, that is the best we can do. However, our simulation results indicate that: (1) narrow pores give higher selectivity and (2) the*

observed selectivity is a result of both pore confinement and cooperative or collective NQEs due to ordering of O₂ molecules when the amount adsorbed is high for preferable pores (close to saturation). These observations are consistent with our hypothesis that narrow pore sites give rise to higher selectivities, because they have higher confinement and are more likely to produce ordered O₂ adsorbed phases than larger nanopores within CDC.

Comment 3.4# Also... a very enigmatic claim of nuclear quantum effects playing a role is brought forward, with no evidence, in particular no benchmark simulation to compare to and no details.

***Response 3.4#** This comment from the reviewer is difficult for us to understand. We have actually included a clear description of the calculations we have carried out with sufficient detail for others to repeat our calculations. Details are given in the supplementary materials. We have carried out state-of-the-art path integral formalism calculations to compute the isotope fractionation ratio for oxygen isotopes showing clearly that the equilibrium selectivity observed at long time in experiments is due to collective quantum effects, not just due to confinement of the sorbent, but to the collective confinement from the co-adsorbed O₂ molecules. This is a new finding and while these values of selectivity are not nearly as high as the transient values at very short times, they are still significantly larger than selectivities of standard processes for oxygen isotope separation.*

Comment 3.5# Finally, the reason for very different time scales is not at all discussed.

***Response 3.5#** The simulation time scales were directed towards equilibrium properties and experimental time scales were focused towards kinetic. It has been included in the revised Supplementary material.*

Comment 3.6# The authors could discuss the applicability to other isotope separation

***Response 3.6#** In order to show the applicability of the present isotope separation method to other isotopes, we have used CDC to demonstrate selective separation of ¹³CH₄ from the mixed gas (¹³CH₄+¹²CH₄) at 112 K. which shows $S(^{13}\text{C} / ^{12}\text{C}) = 56 \pm 6$*

initially and maintains $S > 2.5$ for more than 40 min., shown in Fig. 1h.

Comment 3.7# Line 78 does not make sense, and I do not believe Ref [23] concurs on the term “quantum sieving” though it hints to possible quantum effects in separation processes, which is not entirely the same thing.

Response 3.7# Ref [23] highlight the discoveries that supported the progress of nanofluidics over the past years, so we have modified these lines. More appropriate references are cited in the revised manuscript.

Comment 3.8# Line 117 “the pore widths”. what calculations are you talking about? It seems that your chemical structures are first equilibrated so I don’t see why you would need that. Why do you subtract that value?

Response 3.8# This can be explained with **Figure L6**. Here, H is the physical pore width of the pore measured from the plane of C-atom centers on one wall to the plane of C-atom centers on the opposite wall. Ordinarily, we determine the effective pore width with physical adsorption of N_2 at 77 K or Ar at 87 K. As the molecular probe molecules are adsorbed on the surface of pore wall, the physical adsorption cannot determine H , but H' . Probe molecules are adsorbed even on the dent structure of the pore wall surface and then the effective pore width measured by physical adsorption does not exactly agree with H' . However, one commonly use an approximated relationship given by $H = H' + \Delta$ where $\Delta = 0.335$ nm. Δ is the separation between adjacent graphitic layers of porous carbon (Reference 32).

Langmuir, Vol. 10, No. 12, 1994 4607

Figure L6. *Definition of the physical width, H , and the internal pore width, H' .*

Comment 3.9# I think an explanation of why S decreases should be brought forward early on for the very broad audience. Fig S10 and S17 have layout issues. Fig 3c is a real simulation? Or is a picture view?

Response 3.9# *Referee's suggestions have been taken into consideration on revision. Fig. 3c is a picture view.*

Comment 3.10# The discussion from line 338 lacks clarity and proofs.

Response 3.10# *This work reports the first experimental and simulation evidence for adsorption separation of heavier isotope gases. Several things are yet to be fully understood. However, on the basis of this work, we would expect to open an avenue of research area to study the role of kinetic selectivity for isotopes apart from hydrogen and helium. However, our simulation results indicate that: (1) narrow pores give higher selectivity and (2) the observed selectivity is a result of both pore confinement and cooperative or collective NQEs due to ordering of O_2 molecules when the amount adsorbed is high for preferable pores (close to saturation). These observations are consistent with our hypothesis that narrow pore sites give rise to higher selectivities, because they have higher confinement and are more likely to produce ordered O_2 adsorbed phases than larger nanopores within CDC.*

We hope to be able to report more clear picture for low temperature studies for heavier isotopes separation in the near future. The above line has been modified in the revised manuscript.

REVIEWER COMMENTS

Reviewer #1 (Remarks to the Author):

The authors have addressed all comments satisfactorily, and the manuscript may be published.

Reviewer #2 (Remarks to the Author):

Revised manuscript on "Adsorption Separation of Heavier Isotope Gases in Subnanometer Carbon Pores"

I thank the authors for the through answers, however, still some points are unclear to me.

My key question is still how can the authors determine the **adsorption amount** during the experiment after starting the dosing by opening valves 4, 5 and 7?

In response 2.3# "*we can determine the **adsorption amount** from 0 min.*" No gas is coming out to the mass spectrometer in this first 40 min, therefore, everything is adsorbed, but how much is just speculation, no measurement. In Figure S7d the magnified signal of the MS is shown. $^{16}\text{O}_2$ shows directly a nearly constant low level for over more than 5 min, this is most likely just the higher background of the MS at mass 32, which occurs naturally, whereas, mass 36 has a much lower natural abundance and therefore negligible background.

The question of the adsorbed amount and the mole fraction continues, e.g., the selectivity is given as a function of time for different dosing rates in Figure S8. However, I am still not aware how the selectivity can be calculated according to Eq (1), since the **adsorbed** ratio between the isotopes (mole fractions) is unknown, i.e. not measurable during the dosing. The non-adsorbed gas mixture is detected in the MS, but what is then the meaning of a selectivity directly when the first $^{18}\text{O}_2$ molecules are detected on the outlet from the sample chamber? I think, it is just an artifact of this kind of measurement procedure and the shift with dosing rate in Figure S8 is just a proof of it. In the main text on page 8 the authors added: "*This decrease in the selectivity may be due to decreased availability of the adsorption sites in subsequent course of time.*" This can be certainly true, since during the course of dosing the whole available surface will be covered with adsorbed molecules and more gas is diffusing through afterwards. The equilibrium occupation of the

adsorption sites by $^{16}\text{O}_2$ and $^{18}\text{O}_2$ (molar fraction) depends on the difference in their respective heat of adsorption. The heat of adsorption is governed by the zero point energy and the difference between the two isotopes is roughly proportional to the square root of their mass ratio, here $36/32$ giving 1.06. Therefore, I don't see why this material should possess such a high selectivity between the two oxygen isotopes. Therefore, I cannot understand this statement given in the SI on page 9: "*On passing through the adsorbent kept at low temperature, $^{18}\text{O}_2$ isotope are preferentially adsorbed in the pores of microporous adsorbent.*" What are the unique adsorption sites preferring $^{18}\text{O}_2$? On the other hand, Figure 4i shows narrow pore sites as preferential sites, however, how many of these sites are expected? In this case, the isotopic difference in adsorption in such a confinement depends on the zero point energy and therefore the mass difference as explained above. Again, the expected difference in molar ratio is not very high. The next sentence is unclear, too: "*Unadsorbed isotope mixture gas is trapped in gas reservoir S2 is monitored to determine the molar concentration.*" How is it trapped and why is it not just pumped by P3, which operates continuously during the dosing experiment?

This brings me back to Figure 1a: "*Illustrative model quantitatively comparing cryogenic distillation separation setup with cryogenic adsorption separation method. Distillation towers must be oriented vertically, while adsorption beds can be configured in many ways. The nanoporous adsorbent bed in the adsorption column preferentially adsorbs $^{18}\text{O}_2$ according to the ratio determined by the adsorbent selectivity.*" Even assuming the questionably high selectivity in the first minute is correct, the high selectivity decreases rapidly with time and the shown beds can be only operated for minutes, then have to be closed and refreshed. Therefore, the illustration gives a misleading impression, furthermore, $^{18}\text{O}_2$ will be preferentially adsorbed and not released as the figure shows.

The last sentence in the abstract is strictly not true: "*A collective nuclear-quantum effect difference between ordered $^{18}\text{O}_2$ and $^{16}\text{O}_2$ molecular assemblies confined in subnanometer pores can explain the observed separation and be applicable to other isotopic gases, as seen in our experiments for separation of $^{13}\text{CH}_4$ from $^{12}\text{CH}_4$ with $S = 56$* " I don't see these high affinity of one isotope and the resulting high selectivity explained by the "*collective nuclear-quantum effect difference*". Additionally, the high selectivity numbers, just seen at the very first minute when gas can penetrate the material, are very misleading and apparently only occurring in CDC.

Reviewer #3 (Remarks to the Author):

I have read the revised of the manuscript by Sanjeev Kumar Ujjain et al. I would like to thank the authors for addressing my comments and adding up valuable information to their supplementary. I believe their claims have much gained in strength, are new, and are of general relevance.

However I am strongly convinced that the manuscript in its current stage is very hard to read and therefore will not appeal to a broad audience. Most importantly, the main messages brought forward by the authors are overwhelmed by enumerations of data results. I strongly advise the authors to review their manuscript along these lines :

General comments

- I found a statement made in response to referees very much worthwhile, that is that new modeling should be done to take into account kinetics in NQ simulations. It's obviously a tremendous task, but your experiments clearly show how modeling would be crucial there. I think this statement should be put out in the conclusion.

- line 163-166 seem to say the same thing as lines 160-163 – please consider revising this section by having clearly one paragraph focusing on one result/one message.

- I believe a summary of the experimental section in one sentence stating the main results is needed : namely that adsorption efficacy decreases with temperature, increases with ... etc. This will help to take away guidelines/design rules.

- In general the figures are too crowded. I do not believe Nature Communications has a limit on figure numbers – though I may be mistaken. Please consider how you may simplify e.g. Fig. 4 that has a great number of subpanels into 2 (for example one focusing on the mechanism of adsorption and one on the adsorption isotherms).

Industrial applicability

- line 98 – you clearly mention industrial applicability as a claim of this paper. Therefore industrial applicability should be discussed and recalled more often in the main text.

- The size of your adsorption device on Fig1a would help have the message go through.

- I think a discussion in the main text on the possible adsorption amounts at realistic scales (along the lines of response 2.11) should be in the main text to warrant the industrial application.

- Line 194. Just to make this statement very clear, I think it's good to say that « We have verified that the method is applicable to other gas, by checking (that on other gas it works). »

- A transition is missing line 243. Maybe reconnect to industrial applications here ?

Modeling section

- As an introduction to the modeling section – and also to answer directly questions such as Comment 1.1 in the main text - I would clearly state that as you are studying separation of isotopes, quantum effects are indispensable in simulations. And then narrow down on the specific methods used and why (for instance why equilibrium and why no tunneling).

- 111-116 is still not clear. Consider starting this section by « We determine pore widths by doing this... ». For 0.8nm, 0.7nm and 1.1 did you subtract 0.33 to them already or not – it is not clear ? I think it would be clearer to mention « the average graphite interlayer spacing 0.33 nm » explicitly.

- I don't understand the schematics of fig . 3c. Gas is supposed to come out all the time whereas here it looks like the O₂ molecules are progressing within the slits and that that time delay is responsible for the change in observed selectivity. Which is not what you want to say. Are you showing only the adsorbed molecules here ? why are they not distributed a bit more uniformly within the slit ?

Miscellaneous

- line 152 – I am not sure the abbreviation MS was commented earlier.

- Line 300 – « to induce observable O₂ adsorption » -- you mean in experiments ? please specify in the main text.

**Title: Adsorption Separation of Heavier Isotope Gases in
Subnanometer Carbon Pores**

Sanjeev Kumar Ujjain, Abhishek Bagusetty, Yuki Matsuda, Hideki Tanaka, Preety Ahuja,
Carla de Tomas, Motomu Sakai, Fernando Vallejos-Burgos, Ryusuke Futamura, Irene
Suarez-Martinez, Masahiko Matsukata, Akio Kodama, Giovanni Garberoglio, Yury
Gogotsi, J. Karl Johnson, Katsumi Kaneko

Manuscript Ref. No. NCOMMS-20-20283A

Response to Reviewer's comments:

*At first, we appreciate your consideration of our manuscript (NCOMMS-20-20283A).
We have considered Reviewers' suggestions carefully to make changes in the manuscript
so as to improve our results and figures (Highlighted in green color).
A detailed point-by-point response to the editor and reviewers' comments is attached
below:*

Reviewer #1 (Remarks to the Author):

The authors have addressed all comments satisfactorily, and the manuscript may be published.

Thank you for your kind acceptance, we appreciate for your suggestive comments and precious time.

Reviewer #2 (Remarks to the Author):

Revised manuscript on "Adsorption Separation of Heavier Isotope Gases in Subnanometer Carbon Pores". I thank the authors for the through answers, however, still some points are unclear to me.

Comment 2.1# My key question is still how can the authors determine the **adsorption amount** during the experiment after starting the dosing by opening valves 4, 5 and 7?

In response 2.3# "we can determine the **adsorption amount** from 0 min." No gas is coming out to the mass spectrometer in this first 40 min, therefore, everything is adsorbed, but how much is just speculation, no measurement. In Figure S7d the magnified signal of the MS is shown. $^{16}\text{O}_2$ shows directly a nearly constant low level for over more than 5 min, this is most likely just the higher background of the MS at mass 32, which occurs naturally, whereas, mass 36 has a much lower natural abundance and therefore negligible background.

Response 2.1# The recent adsorption separation experiment data presented as Figure S14 in the previously revised manuscript are performed using CDC adsorbent for only 10 min, 15 min and 20 min at 112 K. They are based on direct determination of the molar ratio of mixture gas in adsorbed phase by desorption. These are kind of breakthrough experiment.

Figure L1. (a) Time courses of mass intensity at room temperature without adsorbent and (b) the mass intensity of desorbed gas for 10 min adsorption at 112 K on CDC.

During these experiments, we performed the adsorption at 112 K for only 10 min, 15 min, 20 min, in the region of high selectivity and determined the adsorbed phase gas amount and their molar ratio. The above figures clearly show that the mass intensity for $^{16}\text{O}_2$ is higher compared to $^{18}\text{O}_2$ in the absence of adsorbent and their molar fraction $^{18}\text{O}_2 : ^{16}\text{O}_2$ in the feed mixture gas before inlet is 1:1. The adsorption was performed using CDC at 112 K for 10 min and the adsorbed amount was determined by desorption at elevated temperature. The desorbed mixture gas 30 Pa, mass intensity of $^{18}\text{O}_2$ is

higher compared to $^{16}\text{O}_2$ and their molar fraction $^{18}\text{O}_2 : ^{16}\text{O}_2 = 1.4 : 1$ (17.5 Pa $^{18}\text{O}_2$ and 12.5 Pa $^{16}\text{O}_2$). This corresponds to a maximum difference of 1.32 mg/g as discussed in the revised manuscript. These are observed/measured values of high reliability. These detailed values are included in the Revised Manuscript-Round 2.

The magnified signal of the MS of Figure S7d shows that $^{18}\text{O}_2$ keeps a nearly constant low level. As per background contribution, a background subtraction is always performed. The near zero mass intensity for $^{16}\text{O}_2$ and $^{18}\text{O}_2$ is at same baseline after background subtraction (black dash circle), as can be evident form Figure S7a. The time courses of mass intensity data without background subtracted is presented in Figure L3 for reference.

Figure L2. Time courses of mass intensity at room temperature and 112 K with or without adsorbent.

Figure L3. Time courses of mass intensity data shown in Fig. L2 a without background subtraction.

Comment 2.2# The question of the adsorbed amount and the mole fraction continues, e.g., the selectivity is given as a function of time for different dosing rates in Figure S8. However, I am still not aware how the selectivity can be calculated according to Eq (1), since the **adsorbed** ratio between the isotopes (mole fractions) is unknown, i.e. not measurable during the dosing. The non-adsorbed gas mixture is detected in the MS, but what is then the meaning of a selectivity directly when the first $^{18}\text{O}_2$ molecules are detected on the outlet from the sample chamber? I think, it is just an artifact of this kind of measurement procedure and the shift with dosing rate in Figure S8 is just a proof of it. In the main text on page 8 the authors added: "*This decrease in the selectivity may be due to decreased availability of the adsorption sites in subsequent course of time.*" This can be certainly true, since during the course of dosing the whole available surface will be covered with adsorbed molecules and more gas is diffusing through afterwards. The equilibrium occupation of the adsorption sites by $^{16}\text{O}_2$ and $^{18}\text{O}_2$ (molar fraction) depends on the difference in their respective heat of adsorption. The heat of adsorption is governed by the zero-point energy and the difference between the two isotopes is roughly proportional to the square root of their mass ratio, here 36/32 giving 1.06. Therefore, I don't see why this material should possess such a high selectivity between the two oxygen isotopes. Therefore, I cannot understand this statement given in the SI on page 9: "*On passing through the adsorbent kept at low temperature, $^{18}\text{O}_2$ isotope are preferentially adsorbed in the pores of microporous adsorbent.*" What are the unique adsorption sites preferring $^{18}\text{O}_2$? On the other hand, Figure 4i shows narrow pore sites as preferential sites, however, how many of these sites are expected? In this case, the isotopic difference in adsorption in such a confinement depends on the zero-point energy and therefore the mass difference as explained above. Again, the expected difference in molar ratio is not very high.

Response 2.2# *Use of Mass spectroscopy (MS) for such adsorption experiment monitoring method is well-established and are used and reported by well-known Adsorption separation groups involving Michael Hirscher et al., Science 2019, 366, 613–620, J. Am. Chem. Soc. 2019, 141, 50, 19850–19858, ChemPhysChem 2019, 20, 1311– 1315 or Andrew I. Cooper et al. Nature Materials 2014, 13, 954–960, J. Am.*

Chem. Soc. 2016, 138, 1653–1659, or Katsumi Kaneko et al. *J. Am. Chem. Soc.* 2012, 134, 18483–18486 and several others. The MS monitors both $^{16}\text{O}_2$ and $^{18}\text{O}_2$ before adsorption and during adsorption, using which the adsorbed phase can be calculated as discussed previously in Supplementary Figure S3 details.

The Figure L2 d, clearly demonstrates that for more than initial 8 minutes, $^{18}\text{O}_2$ are not detected by MS (mass intensity constant), but only $^{16}\text{O}_2$ mass intensity has shown increment from initiation. Selectivity never means, almost all $^{18}\text{O}_2$ molecules should be adsorbed, instead $^{18}\text{O}_2$ are preferentially adsorbed, that's why their mass intensity increment is very slow compared to $^{16}\text{O}_2$.

We must mention again that the established theory for quantum molecular sieving for light isotope molecules cannot be applied to the selective adsorption of heavy isotope molecules such as oxygen and methane isotope molecules. This is already described in Figure S1 in which the Feynman-Hibbs effective potential calculation does not give meaningful differences in the potential minimum and effective size for $^{18}\text{O}_2$ - $^{16}\text{O}_2$ and $^{13}\text{CH}_4$ - $^{12}\text{CH}_4$. Therefore, we cannot apply the concept of the zero-point energy difference to adsorption behavior of these heavier isotope molecules. Coauthor J. Karl, University of Pittsburgh, being a key pioneer of quantum molecular sieving theory for light isotope molecules, and his colleague succeeded to provide a new proof-of-concept that collective nuclear quantum effects are responsible for the long-time (near equilibrium) selectivity observed in experiments. This theory is directly applicable to an equilibrium adsorption, giving the explanation of the observed equilibrium adsorption difference for oxygen isotopes. However, we must wait another challenge for developing a new quantum theory for the observed kinetic molecular sieving effect. This point is also described in the response for Reviewer 3. J.J.Beenakker et al. (*Chem. Phys. Lett.* (1995)) firstly proposed the concept of molecular sieving effect for light isotope molecules. We must wait an established theory for kinetic quantum molecular sieving by S. K. Bhatia et al. (*Phys. Rev. Lett.* (2005)) in case of the quantum molecular sieving for light isotope molecules. As kinetic molecular sieving adsorption processes have more factors than equilibrium selective adsorption, we are looking forward future challenges for this important subject.

Reviewer states here “In this case, the isotopic difference in adsorption in such a confinement depends on the zero-point energy and therefore the mass difference as explained above. Again, the expected difference in molar ratio is not very high”. This statement is based on the quantum molecular sieving based adsorption separation of lighter isotope pairs involving $\text{H}_2/\text{D}_2/\text{T}_2$ etc. As mentioned above, we must understand

the selective adsorption for oxygen and methane isotope molecules observed in this work on the basis of the collective nuclear quantum effects.

The reviewer asks how many of these sites are expected. We described it in the first revised manuscript. The pore structure of CDC having narrow pore sites presented here are unique pore geometry for carbon adsorbents known so-far. They are considered unique adsorption sites because of their in-pore geometry. They preferentially adsorb the heavier $^{18}\text{O}_2$ isotope. As discussed on Page 24 “The cumulative pore volume of such narrow spaces is $V_{\text{narrow-pore}} (d < 0.4 \text{ nm}) = 0.021 \text{ cm}^3 \text{ g}^{-1}$, which contributes to 3.5% of the total pore volume) (Supplementary Figure S24)”.

Comment 2.3# The next sentence is unclear, too: "Unadsorbed isotope mixture gas is trapped in gas reservoir S2 is monitored to determine the molar concentration." How is it trapped and why is it not just pumped by P3, which operates continuously during the dosing experiment?

Response 2.3# The language has been improved to enhance its readability. Yes, it is just pumped by P3. The excess mixture gas if remains after experiment, is stored in reservoir S2. Thank you.

Comment 2.4# This brings me back to Figure 1a: "Illustrative model quantitatively comparing cryogenic distillation separation setup with cryogenic adsorption separation method. Distillation towers must be oriented vertically, while adsorption beds can be configured in many ways. The nanoporous adsorbent bed in the adsorption column preferentially adsorbs $^{18}\text{O}_2$ according to the ratio determined by the adsorbent selectivity." Even assuming the questionably high selectivity in the first minute is correct, the high selectivity decreases rapidly with time and the shown beds can be only operated for minutes, then have to be closed and refreshed. Therefore, the illustration gives a misleading impression, furthermore, $^{18}\text{O}_2$ will be preferentially adsorbed and not released as the figure shows.

Response 2.4# Yes, we agree with the reviewer that “the high selectivity decreases rapidly with time and the shown beds can be only operated for minutes, then have to be closed and refreshed”. That’s why we performed the recent Adsorption separation experiments using CDC adsorbent for only 10 min, 15 min and 20 min at 112 K. Results are included as Figure S14. Based on this we determined the molar ratio of mixture gas

in adsorbed phase by desorption. The adsorbed phase mixture gas (30 Pa), is composed of 17.5 Pa $^{18}\text{O}_2$ and 12.5 Pa $^{16}\text{O}_2$. This corresponds to a maximum difference of 1.32 mg/g as discussed in the revised manuscript. Hence, in summary we have proposed "The highly efficient separation of $^{18}\text{O}_2$ or $^{13}\text{CH}_4$ evidenced in this study can be implemented in industry by designing a rapid adsorption separation process and will facilitate medical and other usage of isotopes". However, we must develop new porous carbon which attains more larger adsorption difference between $^{18}\text{O}_2$ and $^{16}\text{O}_2$. As application of this highly selective separation should conquer several barriers, we remove the conceptual figures in Figure 1a.

Comment 2.5# The last sentence in the abstract is strictly not true: "*A collective nuclear-quantum effect difference between ordered $^{18}\text{O}_2$ and $^{16}\text{O}_2$ molecular assemblies confined in subnanometer pores can explain the observed separation and be applicable to other isotopic gases, as seen in our experiments for separation of $^{13}\text{CH}_4$ from $^{12}\text{CH}_4$ with $S = 56$* " I don't see these high affinity of one isotope and the resulting high selectivity explained by the "*collective nuclear-quantum effect difference*". Additionally, the high selectivity numbers, just seen at the very first minute when gas can penetrate the material, are very misleading and apparently only occurring in CDC.

Response 2.5# *This theory completely explains the equilibrium state not the kinetic, but ordinarily kinetic phenomenon shows concurrence with the equilibrium adsorption. So, just associated the equilibrium state simulation results with the kinetic separations, as explanations on the kinetic separation is still very difficult. Basically, the model should explain the kinetic separation, however the real model is challenging.*

However, we revise the abstract. "A collective nuclear-quantum effect difference between ordered $^{18}\text{O}_2$ and $^{16}\text{O}_2$ molecular assemblies confined in subnanometer pores can explain the observed equilibrium separation and is applicable to other isotopic gases, as seen in our experiments for separation of $^{13}\text{CH}_4$ from $^{12}\text{CH}_4$ with $S = 56 \pm 6$. Additionally, this theory provides a firm foundation for the development of a new quantum kinetic theory for highly selective adsorption of heavy isotope molecules observed in this study."

We can observe similar selective adsorption in several nanoporous materials such as zeolites, single wall carbon nanotube, and activated carbon fiber in addition to CDC, although the reviewer states that --- apparently only occurring in CDC.

The selectivity data shown in Fig.1 for CDC and Fig. 2 for other suitable adsorbents, clearly demonstrates that high selectivities are observed for all adsorbents having a suitable pore geometry. However, the time duration of high selectivity region varies, which shows dependence on the pore size, shape and volume, as mentioned in the preceding Response to the reviewers. In addition, we have also performed Breakthrough experiments using ACFs (data presented in Fig. 3), The high selective adsorption of $^{18}\text{O}_2$ over $^{16}\text{O}_2$ has also been further ascertained by these results.

Reviewer #3 (Remarks to the Author):

I have read the revised of the manuscript by Sanjeev Kumar Ujjain et al. I would like to thank the authors for addressing my comments and adding up valuable information to their supplementary. I believe their claims have much gained in strength, are new, and are of general relevance.

However I am strongly convinced that the manuscript in its current stage is very hard to read and therefore will not appeal to a broad audience. Most importantly, the main messages brought forward by the authors are overwhelmed by enumerations of data results. I strongly advise the authors to review their manuscript along these lines:

***Response:** Thank you very much for kind suggestions and encouragements to improve our manuscript. We revised the manuscript after your suggestions.*

Comment 3.1# I found a statement made in response to referees very much worthwhile, that is that new modeling should be done to take into account kinetics in NQ simulations. It's obviously a tremendous task, but your experiments clearly show how modeling would be crucial there. I think this statement should be put out in the conclusion.

***Response 3.1#** Reviewer's suggestion is quite worth noticing, it has been included conclusion (Highlighted in green color) in the Revised Manuscript-Round 2.*

“However, additional modelling should be performed for the kinetic sieving effect observed in this study for comparison to the NQE simulation for the equilibrium adsorption separation.”

Comment 3.2# Line 163-166 seem to say the same thing as lines 160-163 – please consider revising this section by having clearly one paragraph focusing on one result/one message.

Response 3.2# The lines are improved in the Revised Manuscript-Round 2. The revised lines are as follows: “The high selectivity trend in the initial stage is retained in the temperature range of 100-150 K, as shown in Fig. 1d. However, the S at 1% pore volume filling and the time period for which $S > 2.5$ both decrease with increasing temperature (Fig. 1e).”

Comment 3.3# I believe a summary of the experimental section in one sentence stating the main results is needed: namely that adsorption efficacy decreases with temperature, increases with ... etc. This will help to take away guidelines/design rules.

Response 3.3# A statement highlighting the experimental main result in summary has been included in the Revised Manuscript-Round 2. “we have demonstrated the incredibly high $^{18}\text{O}_2$ adsorption selectivity of nanoporous materials that increases with decreasing temperature and has a strong dependence on the pore geometry of nanoporous materials.”

Comment 3.4# In general the figures are too crowded. I do not believe Nature Communications has a limit on figure numbers – though I may be mistaken. Please consider how you may simplify e.g. Fig. 4 that has a great number of subpanels into 2 (for example one focusing on the mechanism of adsorption and one on the adsorption isotherms).

Response 3.4# As suggested, Fig. 4 has been simplified by placing the magnification figure in Supplementary material.

Fig. 4. Theoretical selectivity and energetics of oxygen isotope adsorption.

Comment 3.5# line 98 – you clearly mention industrial applicability as a claim of this paper. Therefore, industrial applicability should be discussed and recalled more often in the main text. The size of your adsorption device on Fig. 1a would help have the message go through. I think a discussion in the main text on the possible adsorption amounts at realistic scales (along the lines of response 2.11) should be in the main text to warrant the industrial application. Line 194. Just to make this statement very clear, I think it's good to say that « We have verified that the method is applicable to other gas, by checking (that on other gas it works) ». A transition is missing line 243. Maybe reconnect to industrial applications here?

Response 3.5# The suggested point are included in the main text Page no. 15 of Revised Manuscript-Round 2.

“The observed maximum adsorption difference of 1.33 mg/g between $^{18}\text{O}_2$ and $^{16}\text{O}_2$ under the present experimental setup during only 10 min of adsorption separation

appears to be highly promising for dynamic separation. Then, if we increase the sample amount to 1 kg of adsorbent, the available adsorption system can give rise to a preferential $^{18}\text{O}_2$ adsorption of 0.8 L(STP) and will be quite useful. A new type of adsorption separation system with a large sample holder and mixed gas reservoir can be constructed that utilizes the initial large selective adsorption of $^{18}\text{O}_2$; this will enable human society to obtain a promising isotope separation technology, considering that the present $^{18}\text{O}_2$ separation technology using distillation requires 10 months. Furthermore, we confirmed this by performing breakthrough measurements with an adsorption column packed with a comparatively large amount of commercially available ACFs to evaluate the separation performance.”

“The plausible adsorption-separation tower size was estimated for future consideration assuming the ideal conditions with the selectivity remaining at 60 and the adsorption capacity of 15 mmol/g. The assumption of the adsorbent density of 500 kg/m^3 and space velocity = 10 leads to three separation towers with the capacity of 3 m (diameter) \times 1 m (height) producing $^{18}\text{O}_2$ of > 95%.”

Included in Fig 1a and figure caption.

Page no. 12: “In addition, we have verified that this method is applicable to other gaseous heavy isotope pairs involving $^{13}\text{CH}_4$ and $^{12}\text{CH}_4$. The CDC demonstrates selective separation of $^{13}\text{CH}_4$ from the mixed gas ($^{13}\text{CH}_4 + ^{12}\text{CH}_4$) at 112 K with $S(^{13}\text{C}/^{12}\text{C}) = 56 \pm 6$ initially and maintains $S > 2.5$ for more than 40 min (Fig. 1h).”

Comment 3.6# As an introduction to the modeling section – and also to answer directly questions such as Comment 1.1 in the main text - I would clearly state that as you are studying separation of isotopes, quantum effects are indispensable in simulations. And then narrow down on the specific methods used and why (for instance why equilibrium and why no tunneling).

Response 3.6# As suggested we have included these details in the main text on Page no. 22-23 of the Revised Manuscript-Round 2.

“We note that isotopic selectivity is an inherently quantum mechanical phenomenon; we therefore employ path-integral molecular dynamics methods to compute the free energy differences for the gas and adsorbed phases of the different isotopes. In addition, adsorption selectivity is a thermodynamic equilibrium property, so that it is appropriate

to use equilibrium path integral methods. Additionally, we note that tunneling does not contribute in any way to the thermodynamic properties, and therefore does not need to be considered here. The experimental results give kinetic selectivity as a function of time. This quantity is not the true selectivity but rather is related to a breakthrough process of dynamic adsorption. This process cannot be modelled using atomistic path integral methods because it would require the detailed knowledge of the time-dependent pore filling process, including diffusion barriers through the irregularities in the adsorbent materials. Thus, it is only possible to model the equilibrium selectivity. However, this modelling does provide proof-of-concept that collective NQEs are responsible for the long-term (near equilibrium) selectivity observed in the experiments.”

Comment 3.7# 111-116 is still not clear. Consider starting this section by « We determine pore widths by doing this... ». For 0.8nm, 0.7nm and 1.1 did you subtract 0.33 to them already or not – it is not clear? I think it would be clearer to mention « the average graphite interlayer spacing 0.33 nm » explicitly.

Response 3.7# The lines are improved in the Revised Manuscript-Round 2.

Comment 3.8# I don't understand the schematics of Fig. 3c. Gas is supposed to come out all the time whereas here it looks like the O₂ molecules are progressing within the slits and that time delay is responsible for the change in observed selectivity. Which is not what you want to say. Are you showing only the adsorbed molecules here ? why are they not distributed a bit more uniformly within the slit ?

Response 3.8# The schematic shown in Fig. 3a, simply reflects, how isotope gas molecule adsorption phase progresses with time. During initial stage most of the molecules are adsorbed at the pore entrance sites, and in subsequent time they diffuse to inner pore areas. Yes, it reflects only the adsorbed molecules. Such a brief description is included now.

As, we have determined that the pore volume is spatially filling for present study, so the adsorbed molecules distribution shown here is not uniform.

“This reflects that the adsorbed molecules can mostly occupy the pore entrance sites during the initial stage of high selectivity and then diffuse to the inner pore region.”

Comment 3.9# line 152 – I am not sure the abbreviation MS was commented earlier.

Response 3.9# Yes, it is already presented before in line 106.

Comment 3.10# Line 300 – « to induce observable O₂ adsorption » -- you mean in experiments? please specify in the main text.

Response 3.10# The lines are improved in the Revised Manuscript-Round 2.

“However, these pores are too small to induce observable O₂ adsorption experimentally.”

REVIEWERS' COMMENTS

Reviewer #2 (Remarks to the Author):

Second revision of manuscript on “Adsorption Separation of Heavier Isotope Gases in Subnanometer Carbon Pores”

I thank the authors for their additional explanations and additions in the text.

The open question is still, how can one understand the experimentally observed high selectivities in the first minute after gas mixture exposure?

I add my question and their answer of the first rebuttal:

Comment 2.12 # The theoretical calculations yield very low selectivities Table S3 and S4, i.e. near 1 except for the narrow (4,4) SWCNT. Table S3 does not say which selectivity, is it the same as in Table S4? These very low selectivities are in line with the small difference in isosteric heats determined experimentally. However, cannot explain or reproduce the high values obtained by the experiments at ultra-low adsorption amounts.

Response 2.12 # The selectivity is defined in the same way for Tables S3 and S4, as given by Eq. (S2). The values in Table S4 are for experimentally realistic loading of the pore. These values are significantly different from 1, because the current technology uses the selectivity ≈ 1.04 . The calculated values are in line with the experimental values at long time in Fig. 2b. The calculations prove that the experimentally observed selectivity at long times (equilibrium) is due to quantum effects.

And from the second rebuttal:

Comment 2.5 # The last sentence in the abstract is strictly not true: "A collective nuclear-quantum effect difference between ordered $^{18}\text{O}_2$ and $^{16}\text{O}_2$ molecular assemblies confined in subnanometer pores can explain the observed separation and be applicable to other isotopic gases, as seen in our experiments for separation of $^{13}\text{CH}_4$ from $^{12}\text{CH}_4$ with $S = 56$ " I don't see these high affinity of one isotope and the resulting high selectivity explained by the "collective nuclear-quantum effect difference". Additionally, the high selectivity numbers, just seen at the very first minute when gas can penetrate the material, are very misleading and apparently only occurring in CDC.

Response 2.5 # This theory completely explains the equilibrium state not the kinetic, but ordinarily kinetic phenomenon shows concurrence with the equilibrium adsorption. So, just associated the equilibrium state simulation results with the kinetic separations, as explanations on the kinetic separation is still very difficult. Basically, the model should explain the kinetic separation, however the real model is challenging.

However, we revise the abstract. "A collective nuclear-quantum effect difference between ordered $^{18}\text{O}_2$ and $^{16}\text{O}_2$ molecular assemblies confined in subnanometer pores can explain the observed equilibrium separation and is applicable to other isotopic gases, as seen in our experiments for separation of $^{13}\text{CH}_4$ from $^{12}\text{CH}_4$ with $S = 56 \pm 6$. Additionally, this theory provides a firm foundation for the development of a new quantum kinetic theory for highly selective adsorption of heavy isotope molecules observed in this study."

For me the question remains open, since, at the moment, the theory of a "collective nuclear-quantum effect difference" only reproduces the equilibrium values. Furthermore, I trust Karl Johnson, that *However, we must wait another challenge for developing a new quantum theory for the observed kinetic molecular sieving effect.*

Let's hope that this publication initiates "the development of a new quantum kinetic theory for highly selective adsorption of heavy isotope molecules" as the authors added to the abstract.

Reviewer #3 (Remarks to the Author):

I have read the revised version of the manuscript. I would like to thank the authors for their continuous effort in making the manuscript clearer and more detailed.

I have only minor comments at this stage. I recommend the publication in Nature Communications; this is a quite impressive technology indeed.

- line 205 - there are no units for the velocity; please specify

- Figure 3 - c - in the caption, following our discussion I would strongly suggest adding a note that the red and blue molecules represent adsorbed molecules.

- line 307 - I would suggest adding before "Details" the blue section highlighted from 351; in fact this blue section is more methodological and justifies the technique used. Hence it does make sense for it to come before the specific results.

**Title: Adsorption Separation of Heavier Isotope Gases in
Subnanometer Carbon Pores**

Sanjeev Kumar Ujjain, Abhishek Bagusetty, Yuki Matsuda, Hideki Tanaka, Preety Ahuja,
Carla de Tomas, Motomu Sakai, Fernando Vallejos-Burgos, Ryusuke Futamura, Irene
Suarez-Martinez, Masahiko Matsukata, Akio Kodama, Giovanni Garberoglio, Yury
Gogotsi, J. Karl Johnson, Katsumi Kaneko

Manuscript Ref. No. NCOMMS-20-20283C

Response to Reviewer's comments:

At first, we appreciate your consideration of our manuscript (NCOMMS-20-20283C). We have considered Reviewers' suggestions carefully to make changes in the manuscript so as to improve our results and figures (Highlighted in green color).

A detailed point-by-point response to reviewers' comments is attached below:

Reviewer #2 (Remarks to the Author):

Second revision of manuscript on "Adsorption Separation of Heavier Isotope Gases in Subnanometer Carbon Pores"

I thank the authors for their additional explanations and additions in the text.

The open question is still, how can one understand the experimentally observed high selectivities in the first minute after gas mixture exposure?

For me the question remains open, since, at the moment, the theory of a "collective nuclear-quantum effect difference" only reproduces the equilibrium values. Furthermore, I trust Karl Johnson, that *However, we must wait another challenge for developing a new quantum theory for the observed kinetic molecular sieving effect.*

Let's hope that this publication initiates "the development of a new quantum kinetic theory for highly selective adsorption of heavy isotope molecules" as the authors added to the abstract.

Response: Thank you for your kind words regarding the work, we appreciate for your suggestive comments and precious time.

Reviewer #3 (Remarks to the Author):

I have read the revised version of the manuscript. I would like to thank the authors for their continuous effort in making the manuscript clearer and more detailed.

I have only minor comments at this stage. I recommend the publication in Nature Communications; this is a quite impressive technology indeed.

- line 205 - there are no units for the velocity; please specify
- Figure 3 - c - in the caption, following our discussion I would strongly suggest adding a note that the red and blue molecules represent adsorbed molecules.
- line 307 - I would suggest adding before "Details" the blue section highlighted from 351; in fact this blue section is more methodological and justifies the technique used. Hence it does make sense for it to come before the specific results.

Response: Thank you very much for kind suggestions and encouragements to improve our manuscript. We have revised the manuscript and included your suggestions in the Final Revised Manuscript.